# Introducing the Dendrify framework for incorporating dendrites to spiking neural networks

Michalis Pagkalos [1,2], Spyridon Chavlis [1] & Panayiota Poirazi [1] ✉

Computational modeling has been indispensable for understanding how subcellular neuronal features influence circuit processing. However, the role of dendritic computations in network-level operations remains largely unexplored. This is partly because existing tools do not allow the development of realistic and efficient network models that account for dendrites. Current spiking neural networks, although efficient, are usually quite simplistic, overlooking essential dendritic properties. Conversely, circuit models with morphologically detailed neuron models are computationally costly, thus impractical for large-network simulations. To bridge the gap between these two extremes and facilitate the adoption of dendritic features in spiking neural networks, we introduce Dendrify, an open-source Python package based on Brian 2. Dendrify, through simple commands, automatically generates reduced compartmental neuron models with simplified yet biologically relevant dendritic and synaptic integrative properties. Such models strike a good balance between flexibility, performance, and biological accuracy, allowing us to explore dendritic contributions to network-level functions while paving the way for developing more powerful neuromorphic systems.

Simulations of spiking neural networks (SNNs) are widely used to understand how brain functions arise from area-specific network dynamics[1–3]. Moreover, SNNs have recently gained much attention for their value in low-power neuromorphic computing and practical machine learning applications[4,5]. SNNs typically comprise point, integrate-and-fire (I&F) neurons and replicate basic biological features such as specific connectivity motifs, excitation-inhibition dynamics, and learning via synaptic plasticity rules[6–9]. However, SNNs often ignore dendrites, the thin membranous extensions of neurons that receive the vast majority of incoming inputs. Numerous studies have shown that the dendrites of excitatory and inhibitory neurons possess compelling computational capabilities[10,11] that can significantly influence both neuronal and circuit function[12–15] and cannot be captured by point-neuron SNNs (for a recent review, see ref. [16]).

First, dendrites can act as semi-independent thresholding units, producing local regenerative events termed dendritic spikes (dSpikes).

These spikes are generated by local voltage-gated mechanisms (e.g., $Na^+/Ca^{2+}$ channels and NMDA receptors) and influence synaptic input integration and plasticity[10,11]. Moreover, dendritic mechanisms operate in multiple timescales, ranging from a few up to hundreds of milliseconds, allowing complex computations, including coincidence detection, low-pass filtering, input segregation/amplification, parallel nonlinear processing, and logical operations[17–22].

Due to these nonlinear phenomena, the arrangement of synapses along dendrites becomes a key determinant of local and somatic responses. For example, the impact of inhibitory pathways depends on their exact location relative to excitatory inputs[23,24]. Moreover, functionally related synapses can form anatomical clusters, which facilitate the induction of dSpikes, thus increasing computational efficiency and storage capacity[25–27]. Finally, dendritic morphology and passive properties shape the general electrotonic properties of neurons[10]. For example, dendritic filtering affects both the amplitude and the kinetics

[1]Institute of Molecular Biology and Biotechnology (IMBB), Foundation for Research and Technology Hellas (FORTH), Heraklion 70013, Greece. [2]Department of Biology, University of Crete, Heraklion 70013, Greece. ✉e-mail: poirazi@imbb.forth.gr

of synaptic currents traveling toward the soma in a location-dependent manner. Given the complexity of dendritic processing, SNNs that lack dendrites may fail to account for important dendritic contributions to neuronal integration and output, limiting their true computational power.

Conversely, biophysical models of neurons with a detailed morphology are ideal for studying how dendritic processing affects neuronal computations at the single-cell level[16]. Such models comprise hundreds of compartments, each furnished with numerous ionic mechanisms to faithfully replicate the electrophysiological profile of simulated neurons. However, achieving high model accuracy is typically accompanied by increased complexity (e.g., higher CPU/GPU demands and larger run times), as numerous differential equations have to be solved at each simulation time step[16]. Therefore, this category of models is unsuitable for large-network simulations, where computational efficiency is a key priority.

A middle-ground solution utilizes simplified models that capture only the essential electrophysiological characteristics of real neurons[28–33]. Notable examples of this approach are found in recent theoretical studies showing that dendritic mechanisms convey significant advantages to simplified network models of varying levels of abstraction. These include improved associative learning[12,27], better

input discrimination (pattern separation[34]), efficient binding/linking of information[12,35], and increased memory storage and recall capacity[14,36]. Similar advantages were recently seen in the machine learning field: adding dendritic nodes in artificial neural networks (ANNs) reduced the number of trainable parameters required to achieve high-performance accuracy[37] (also see [38,39]). Moreover, incorporating dendritic nodes in Self Organizing Map classifiers[40] and other neuro-inspired networks[41] improved their continuous learning ability.

Overall, while dendrites confer advanced computational power to simulated biological networks and these benefits are likely to extend to machine learning systems, SNNs remain largely dendrite-ignorant. A likely reason is that the current theoretical framework for modeling dendritic properties consists of overly complex equations with numerous free parameters, making it mathematically intractable and impractical for use in SNNs.

To address the abovementioned complexity issues and provide a framework that allows the seamless incorporation of dendrites in SNN models, we developed Dendrify (Fig. 1). Dendrify is a free, open-source Python package that facilitates the addition of dendrites and various dendritic mechanisms in SNNs. Importantly, Dendrify works with the Brian 2 simulator[42]; it builds upon the latter's powerful and flexible features while automating some potentially complex and error-prone steps related to

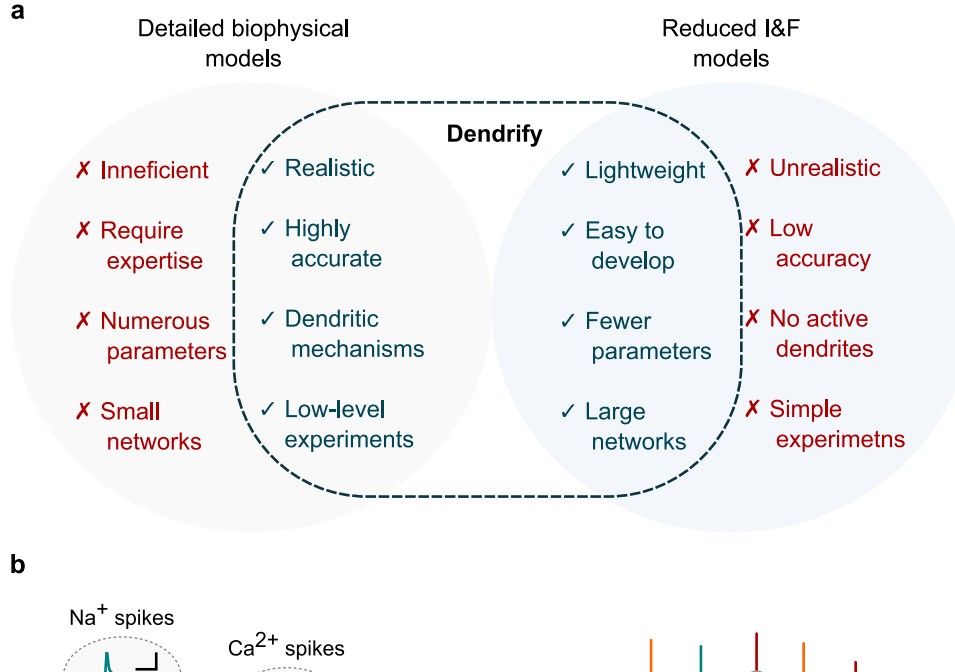

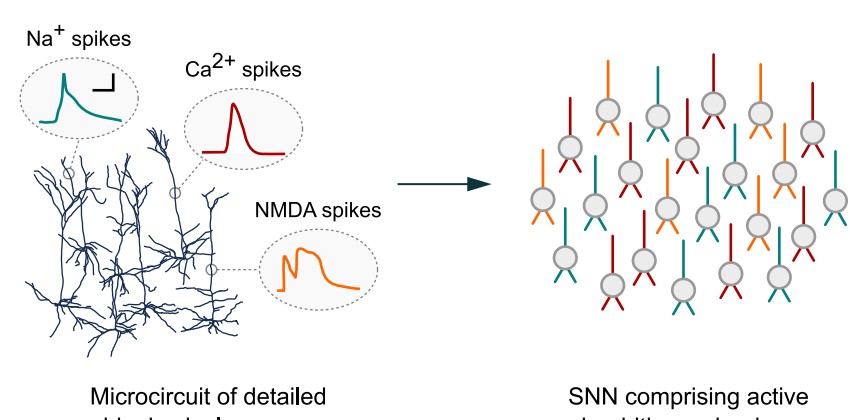

**Fig. 1 | The main characteristics of dendrify. a** Dendrify stemmed from our efforts to bridge the gap between detailed biophysical models and reduced I&F models. The result is a modeling framework for developing simplified compartmental models that balance efficiency and biological accuracy by capturing the most important characteristics of both worlds. **b** Dendrify facilitates the development of SNNs comprising reduced compartmental neurons (ball and sticks) and known dendritic phenomena, such as various types of local spikes (Color code; teal: Na⁺ spikes, red: Ca²⁺ spikes, orange: NMDA spikes. Scale bar: 20 mV/10 ms).

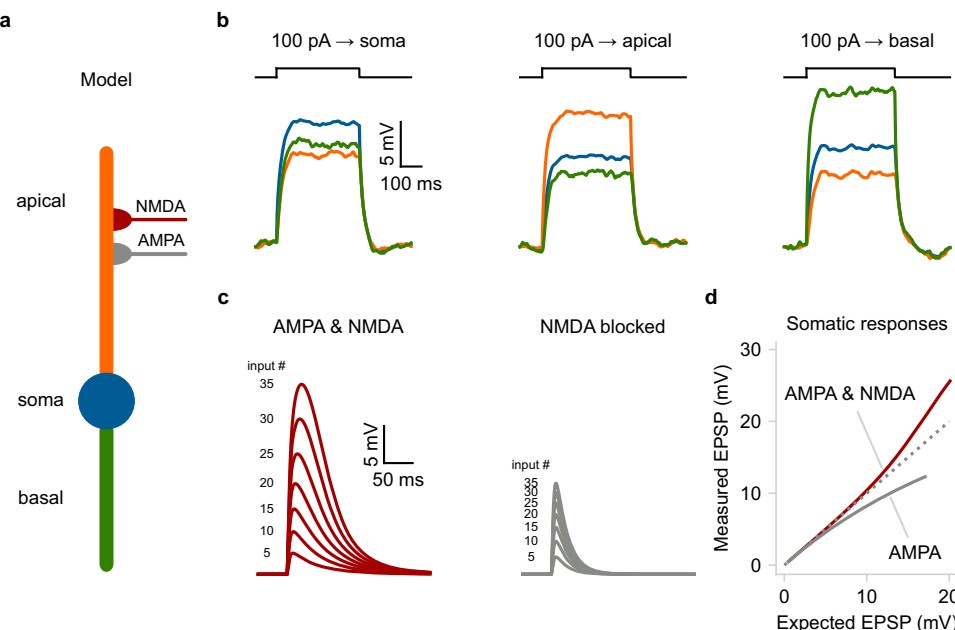

**Fig. 2 | A basic compartmental neuron model with passive dendrites.**
**a** Schematic illustration of a compartmental model consisting of a soma (spiking unit) and two dendrites (passive integrators). The apical dendrite can integrate excitatory synapses comprising AMPA and NMDA currents. **b** Membrane voltage responses to current injections of the same amplitude are applied individually to each compartment. Notice the electrical segregation caused by the resistance between the three neuronal compartments. **c** Somatic responses to a varying number of simultaneous synaptic inputs (5–35 synapses). Left: control EPSPs, right: EPSPs in the presence of NMDA blockers. d) Input-output function of the apical dendrite as recorded at the soma. The dotted line represents a linear function. Notice the shift from supralinear to sublinear mode when NMDARs are blocked. The simulations and analysis code related to the above figure can be executed in any browser by following this link: https://github.com/Poirazi-Lab/dendrify/blob/main/paper_figures/Fig2_notebook.ipynb.

compartmental modeling. Specifically, through simple and intuitive commands, Dendrify automatically generates and handles all the equations (and most parameters) needed by Brian 2 to build simplified compartmental neurons. Its internal library of premade models supports a broad range of neuronal mechanisms yet allows users to provide their own model equations. Among other optimizations, we also introduce a novel phenomenological approach for modeling dSpikes, significantly more efficient and mathematically tractable than the Hodgkin–Huxley formalism. Moreover, we provide a step-by-step guide for designing reduced compartmental models that capture the key electrophysiological and anatomical properties of their biological counterparts. Notably, the proposed guide builds upon established theoretical work[28,29,31], and its implementation is not exclusive to any simulator software. To our knowledge, this is the first systematic approach that combines a theoretical framework with a tool for adding dendrites to simple, phenomenological neuronal models in a standardized and mathematically concise manner.

## Results

To demonstrate the power of Dendrify, we showcase its main features through four modeling paradigms of increasing complexity. (a) A basic compartmental model with passive dendrites, (b) a reduced compartmental model with active dendrites, (c) a simplified model of a CA1 pyramidal neuron that reproduces numerous experimental observations, and d) a pool of CA1 neurons used to assess the contribution of dendritic Na+ spikes in coincidence input detection. In addition, to demonstrate Dendrify's scalability and low computational cost, we compare the execution time for both single-cell and network models of increasing complexity and size.

### Example 1: A basic compartmental model with passive dendrites

We start with a simple neuron model consisting of three compartments (Fig. 2a). A soma, modeled as a leaky I&F unit, and two passive

dendrites (apical and basal) that are electrically coupled to the soma (see Methods). This architecture roughly resembles the general dendritic organization of excitatory, pyramidal-like neurons. In this example, the apical dendrite can integrate excitatory synaptic inputs consisting of a fast α-amino-3-hydroxy-5-methyl-4-isoxazolepropionic acid (AMPA) component and a slow N-methyl-D-aspartate (NMDA) component. In addition, both dendritic compartments are connected to a source of Gaussian white noise (i.e., noisy input current). The Python code needed to reproduce this model is described in Supplementary Fig. 1. All model parameters are available in Supplementary Table 1.

To test our model's electrical behavior, we applied depolarizing current injections (400 ms pulses of 100 pA at −70 mV baseline voltage) individually to each compartment and recorded the voltage responses of all compartments (Fig. 2b). As expected, the largest depolarization was observed at the current injection site, while compartments located further apart were less affected, demonstrating the model's ability to capture the signal attenuation features of biological neurons[10,43]. Note that the basal dendrite in this model is more excitable than the apical one due to the difference in length (150 μm vs. 250 μm, respectively). The attenuation of currents traveling along the somatodendritic axis is an intrinsic property of biological neurons and is due to the morphology and cable properties of dendritic trees[10,43].

Although dendritic attenuation may seem undesirable, it has several computational advantages[10]. For instance, it allows dendrites to operate semi-independently from the soma[44] and perform complex functions, especially when paired with local voltage-gated mechanisms. In our toy model, simultaneous activation of an increasing number of synapses on the apical dendrite evokes somatic responses much larger than the expected arithmetic sum of individual inputs (Fig. 2c, d). The additional depolarization is due to the activation of NMDARs (at elevated dendritic voltages), resulting in supralinear integration. However, when NMDARs are

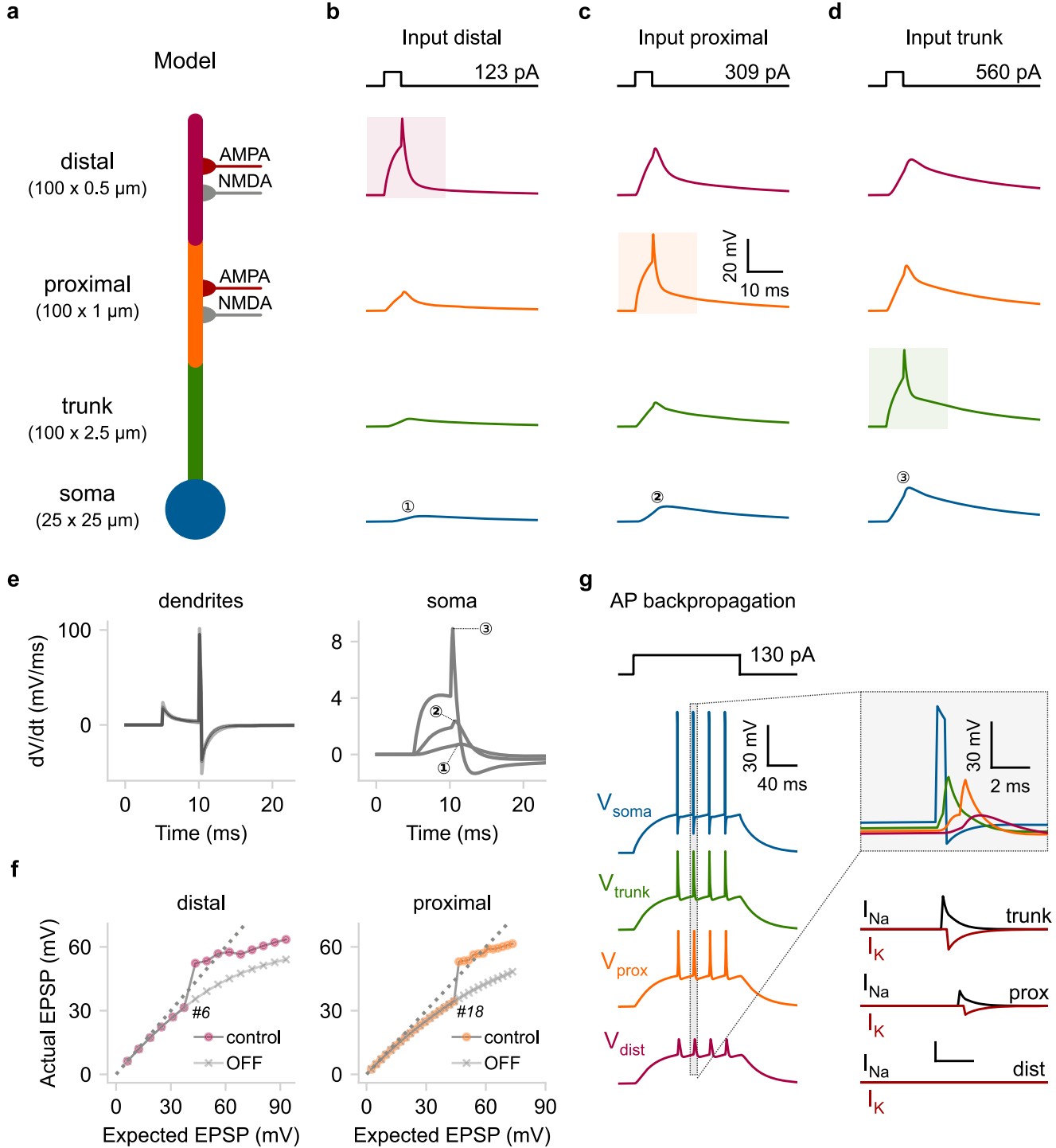

**Fig. 3 | A reduced compartmental model that replicates active dendritic properties. a** Schematic illustration of a compartmental model consisting of a soma (leaky I&F) and three dendritic segments (trunk, proximal, distal) equipped with Na⁺-type VGICs. The distal and proximal segments can also receive AMPA and NMDA synapses. **b–d** Rheobase current injections (5 ms square pulses) for dSpike generation were applied individually to each dendritic segment. Shaded areas: location of current injection and dSpike initiation. Top: stimulation protocol showing the current threshold for a single dSpike (rheobase current). **e** First temporal derivative of dendritic (left) and somatic (right) voltage traces from panels (**b–d**). **f** Input–output function of the distal (left) and proximal (right) segment as recorded from the corresponding dendritic locations. We also indicate the number of quasi-simultaneously activated synapses (ISI = 0.1 ms) needed to elicit a single dSpike in each case. OFF: deactivation of Na⁺ dSpikes. Dashed lines: linear input–output relationship. **g** Left: Backpropagating dSpikes are generated in response to somatic current injections. The short-amplitude spikelets detected in the distal branch are subthreshold voltage responses for dSpike initiation. Right: Magnified and superimposed voltage traces (top) from the dashed box (left). Below: dendritic voltage-activated currents responsible for dSpikes generation in each dendritic segment. The simulations and analysis code related to the above figure can be executed in any browser by following this link: https://github.com/Poirazi-Lab/dendrify/blob/main/paper_figures/Fig3_notebook.ipynb.

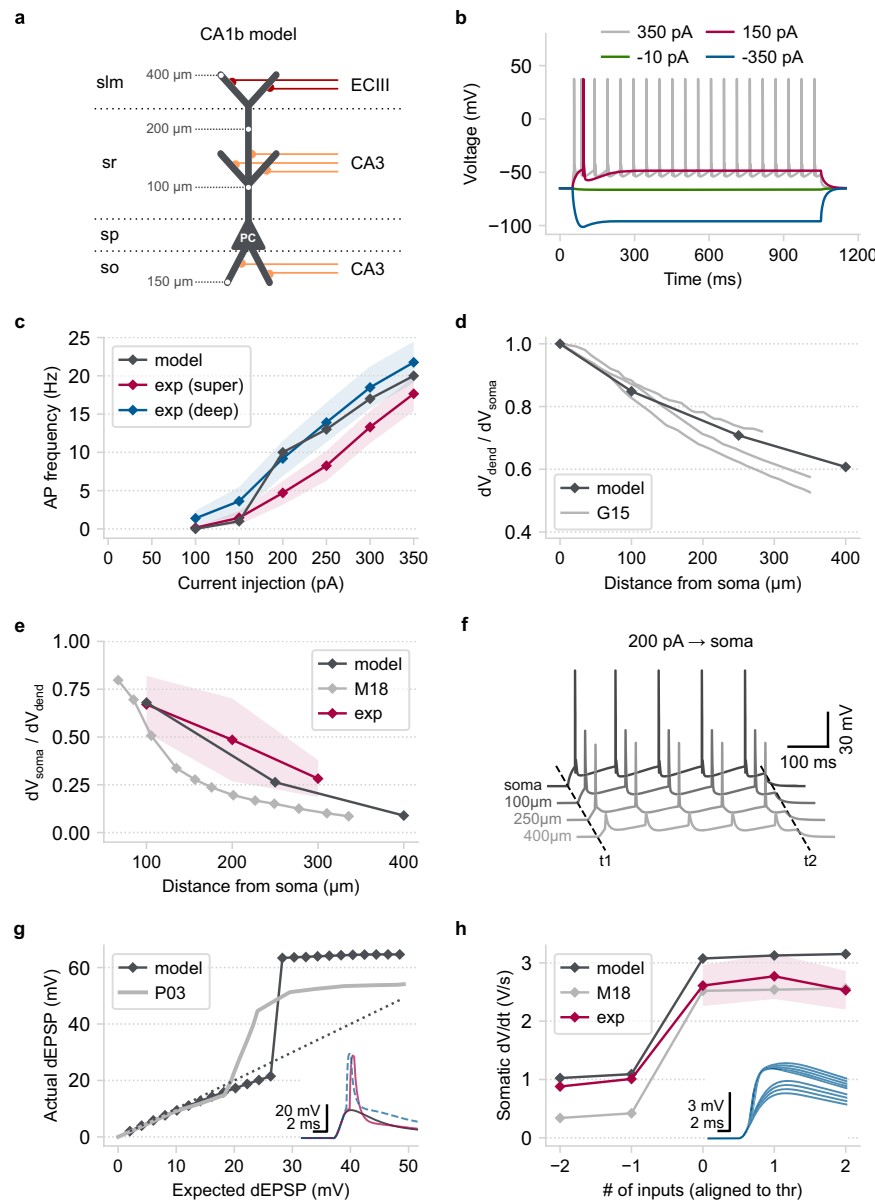

**Fig. 4 | CA1 pyramidal model validation. a** Schematic illustration of the reduced CA1 PC model consisting of a somatic and eight dendritic segments (2× basal, 1× proximal trunk, 1× distal trunk, 2× radial oblique, 2× distal tuft). Grey numbers: distance of the indicated points from the soma. Red axons: EC layer III input, orange axons: CA3 input. Horizontal dotted lines: borders of the four CA1 layers (slm: stratum lacunosum-moleculare, sr: stratum radiatum, sp: stratum pyramidale, so: stratum oriens). **b** Somatic voltage responses to various (1000 ms long) current injections used for model validation. **c** $F$–$I$ curves comparing the model with actual superficial and deep PCs located in the CA1b area[54]. Shaded area: SEM. **d** Steady-state, distance-dependent voltage attenuation of a long current pulse injected at the soma. G15: data for three detailed biophysical models adapted from[55]. **e** The attenuation of postsynaptic currents propagating along the apical dendrite as a function of distance from the soma. M18: biophysical modeling data adapted from[67], Exp: experimental data adapted from[102]. Shaded area: two standard deviations. **f** Simultaneous somatodendritic recordings in response to a somatic current injection showing the emergence of BPAPs. T1/T2: start/end of current injection (duration = 500 ms). **g** Main panel: Input-output function of the reduced model's oblique dendrite (the interval between inputs is 0.1 ms). P03: biophysical modeling data adapted from[44]. Arrows: indicate a different number of co-active synapses (grey = 13, pink = 14, blue = 24). Inset: dendritic voltage responses from the three highlighted cases. **h** Main panel: peak d$V$/d$t$ of somatic voltage responses as a function of synaptic inputs (data aligned to their respective thresholds for dSpike initiation). M18: biophysical modeling data adapted from[67]. Exp: experimental data adapted from[102]. Shaded areas: SEM. Inset: First temporal derivative of the reduced model's somatic EPSPs. Numbers indicate the number of co-active synapses on the apical oblique dendrites. The simulations and analysis code related to the above figure can be executed in any browser by following this link: https://github.com/Poirazi-Lab/dendrify/blob/main/paper_figures/Fig4_notebook.ipynb.

blocked, the apical dendrite switches from supralinear to a sublinear integration mode (Fig. 2c, d), and this alteration can be dendrite-specific. This happens because synaptic currents are susceptible to the decrease in driving force as dendritic voltage approaches the AMPA reversal potential ($E_{AMPA} = 0$ mV). Both types of dendritic integration have been observed in real neurons and allow distinct computations, such as e.g. clustered vs. scattered input sensitivity[43].

This example shows that even rudimentary compartmental models can simulate essential dendritic functions like signal attenuation and segregation that point-neuron models cannot capture. Importantly, they allow the presence of multiple input segregation sites, theoretically enhancing the computational capacity of single neurons[45]. In addition, we provide an example of how even basic dendritic-driven mechanisms can impact neuronal integration and somatic output.

## Example 2: A reduced compartmental model with active dendrites

In the previous example, dendrites were modeled as passive leaky compartments with added synaptic mechanisms. However, a unique feature of Dendrify is the ability to incorporate voltage-gated ion channels (VGICs, see Methods) that are implemented phenomenologically without utilizing the Hodgkin–Huxley formalism. This approach further reduces mathematical and computational complexity, as exemplified by a second reduced model (parameters shown in Supplementary Table 2) consisting of a somatic compartment (leaky I&F) and an apical dendrite divided into three segments (Fig. 3a, Supplementary Fig. 2). All dendritic compartments are equipped with models of $Na^+$-type VGICs (allowing the generation of $Na^+$ dSpikes), while the distal and proximal segments can integrate synaptic inputs consisting of AMPA and NMDA currents.

First, to test the impact of locally generated $Na^+$ spikes on dendritic and somatic responses in the model neuron, we simulated the application of short current injections (5 ms long pulses of rheobase intensity) to each dendritic segment and recorded simultaneously from all compartments (Fig. 3b–d). Although model parameters were adjusted to elicit nearly identical responses in all dendritic compartments (Fig. 3e left), somatic responses varied significantly, depending on the dSpike initiation site (Fig. 3e right). As in real neurons, distal dSpikes became much weaker and broader as they traveled toward the soma due to the dendritic filtering effect[10,46].

Moreover, the threshold for dendritic spiking significantly differs among the three dendritic locations (Fig. 3b–d top). For example, dSpike generation in the distal segment (Fig. 3b) requires approximately 2.5 times less current than the proximal one (Fig. 3c). Due to its smaller diameter and sealed end, the distal segment has higher input resistance ($R_{input}$); thus, its membrane is more excitable. Biological neurons also exhibit a large variability of axial resistance along their dendrites caused mainly by differences in local branch dimensions (length and diameter) and dendritic geometry (e.g., bifurcations number and branch order). This location-dependent change in input resistance (typically increases in the thinner, distal branches) serves two functions. First, it increases the probability of dSpike initiation in the distal dendritic branches, and second, it helps to counterbalance the distance-dependent input attenuation caused by cable filtering[43,46,47].

To examine how dendritic spiking combined with local branch properties affect synaptic integration in our toy model, we activated quasi-simultaneously (train of spikes with ISI 0.1 ms) an increasing number of synapses placed on the distal and the proximal segments. We then compared the peak amplitude of the dendritic voltage responses (Actual) to what would be obtained by a linear arithmetic sum of unitary responses (Expected) (Fig. 3f). Both segments produce voltage responses that increase in a sigmoid-like fashion, with a supralinear rise in their amplitude occurring above a certain number of synapses (Fig. 3f control). This behavior is typical of pyramidal neurons in the cortex and the hippocampus[17,19,44,48], as well as some interneurons[13,49]. Moreover, blocking dSpikes (Fig. 3f OFF) disrupts the above response leading to sublinear integration. Although the two segments appear to have similar input-output curves, dendritic nonlinearities emerge earlier in the distal compartment. This is because of its higher input resistance ($R_{input}$), requiring less synaptic excitation to cross the dSpike voltage threshold. This model property, which is based on experimental data[46], highlights the importance of accounting for input pathways projecting to different dendritic locations, as they may be subject to different integration rules. Notably, the same approach used to build phenomenological models of $Na^+$ dSpikes can be used to build models of other types of local-generated spikes (e.g., $Ca^{2+}$-based).

Another key feature of biological neurons is the ability of APs initiated in the axon to invade the soma and nearby dendrites and propagate backward toward the dendritic tips. The transmission efficacy of these backpropagating action potentials (BPAPs) depends on the dendritic morphology and the abundance of dendritic VGICs ($Na^+$ or $Ca^{2+}$)[45]. Notably, in several neuronal types, BPAPs can propagate more efficiently than forward-propagating dSpikes, acting as feedback signals of somatic activity[45] and serving as instructive plasticity signals[50–52]. To test if our model can recreate the generation of BPAPs, we injected a depolarizing step current at the soma (135 pA for 300 ms) capable of eliciting a small number of somatic APs (Fig. 3f). Upon somatic activation (the axon is not explicitly modeled here), BPAPs were successfully generated and propagated to the distal dendritic segment. There, dSpikes were reduced to sharp, small-amplitude responses (spikelets), as often observed experimentally[53]. These spikelets resulted from attenuating ion influxes from nearby dSpikes, that failed to trigger local suprathreshold responses. It should be noted that to achieve BPAP generation, we had to utilize a custom version of the I&F model[27] that results in a more realistic somatic AP shape (see Methods).

Altogether, the above simulations show that Dendrify allows the development of reduced compartmental models that incorporate phenomenological voltage-gated mechanisms and can replicate various dendritic features and their impact on somatic output. These reduced yet more biologically relevant models offer a compelling alternative for developing SNNs with a high degree of bioinspiration and small computational overhead. Importantly, Dendrify provides easy access to this category of models by radically simplifying their implementation in Brian 2.

## Example 3: A simplified yet biologically accurate model of a CA1 pyramidal cell

The previous examples demonstrated how Dendrify promotes the development of simple compartmental models reproducing several essential dendritic functions. However, our examples comprised generic neuron models rather than any area-specific cell type. To explore our approach's full potential and limitations, we built a simplified yet realistic model of a CA1 pyramidal cell (PC). This cell type was selected due to the availability of a large volume of experimental data[54] and computational models[55,56] to compare our work with. To keep our approach simple, we did not use third-party software to design the model's morphology[57] or fit its parameters[58]. Instead, based on previous theoretical work[28,29,31], we created a set of instructions that guides Dendrify users throughout model development and validation processes. The specific approach is briefly discussed below (for a more detailed description, see Methods).

Our reduced CA1 PC model (Fig. 4a) consists of 9 segments (1 somatic + 8 dendritic), the dimensions of which were constrained using mouse anatomical data[59,60]. All model parameters are provided in Supplementary Table 3. Our goal was to preserve: (a) the main functional and anatomical characteristics of the dendritic morphology, (b) compartment-specific synaptic placement, and (c) realistic dendritic attenuation (axial resistance). In particular, this morphology reflects the anatomical layering of the CA1 hippocampal area and the spatial segregation of input pathways coming from the entorhinal cortex (EC) and the CA3 area. Moreover, synaptic conductances were manually calibrated to ensure that the AMPA to NMDA ratio and the unitary postsynaptic responses along the dendritic tree agree with empirical data (Supplementary Fig. 4, Supplementary Table 3)[61–66]. To directly compare our model with the available in vitro data[54], we replicated the experimental procedures used to estimate essential electrophysiological properties (Fig. 4b, c, Supplementary Fig. 3). We observe that the model's membrane time constant ($\tau_m$), input resistance ($R_{input}$), sag ratio, and F–I curve closely approximate the respective properties of real PCs located in the CA1b subregion, the most central part of the CA1 area.

Since studies with simultaneous somatodendritic recordings are scarce in the literature, we utilized data from various sources

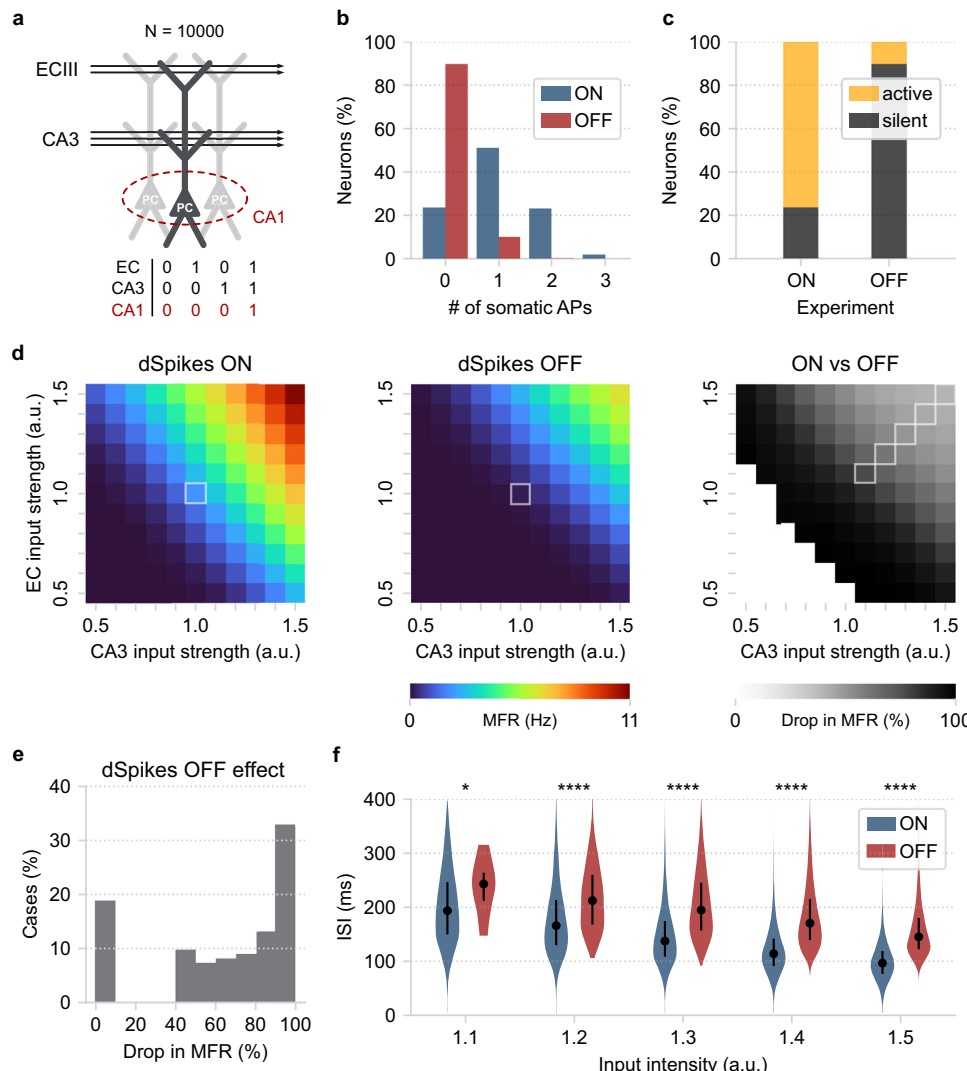

**Fig. 5 | Pathway interaction in a reduced CA1 network model. a** Schematic illustration of a pool of reduced compartmental CA1 PCs ($N = 10,000$). The arrows represent the two streams of input (independent Poisson-distributed spike trains) projecting to distinct dendritic segments. Each neuron represents a repetition of the same experiment with independent Poisson-distributed inputs of the same average frequency. Bottom: table describing the conditional activation of CA1 PCs requiring coincident EC and CA3 input. **b** Probability distribution of somatic spike count, with (ON) or without (OFF) dendritic spikes, when both EC and CA3 input is applied to the network. **c** Summary of the results shown in panel (**b**). Active neurons: PCs that fired ≥1 somatic spike. Notice the reduction of the active population size when dendritic spiking is turned off. **d** Repeating the coincidence detection experiment for a broad range of input intensities. Left: Mean neuronal firing rate (MFR) for each combination of EC/CA3 input amplitudes. Centre: same as in Left but with dSpikes turned off. The highlighted squares indicate the initial

experimental conditions for the data shown in panels (b, c. Right: quantifying the decrease in coincidence detection efficacy by measuring the MFR percentage decrease (dSpikes ON vs. dSpikes OFF). Deactivation of dendritic spiking results in reduced MFR in all cases tested. The white squares (bottom left) represent cases with very low initial MFR (<0.1 Hz or <5% network activity) that were excluded from the analysis. The highlighted squares indicate the experimental conditions of the data shown in panel (**f**). **e** Distribution of the results shown in panel (**d**) (right). **f** Comparing the ISI distributions between the dSpikes ON and OFF conditions, using the highlighted cases in panel (**d**) (right). The circles represent the distribution medians, and the vertical lines are the first and third quantiles containing 50% of the data. Stars denote significance with unpaired t-test (two-tailed) with Bonferroni's correction. The simulations and analysis code related to the above figure can be executed in any browser by following this link: https://github.com/Poirazi-Lab/dendrify/blob/main/paper_figures/Fig5_notebook.ipynb.

(experimental[48,61] and modelling[44,55,56,67]) to calibrate our model's dendritic properties. First, to quantify dendritic attenuation as a function of distance from the soma, we injected current at the soma (1000 ms square pulse of −10 pA) and calculated the ratio of the dendritic to somatic steady-state voltage responses ($dV_{dend}/dV_{soma}$) at various locations. The reduced model is similar to three detailed biophysical models[55] (Fig. 4d). Next, to examine synaptic input attenuation, we activated synapses (single pulse with a time interval of 0.1 ms) at various apical dendrite locations and calculated the somatic to dendritic peak voltage ($dV_{soma}/dV_{dend}$) (Fig. 4e). Compared to experimental data[61] and a recent, highly optimized biophysical model[67], the reduced model captures the distance-dependent attenuation of EPSPs. It

should be noted that the high variability in the morphology[60] and the electrophysiological properties[54] of real CA1 PCs make any attempt to build a single (detailed or simplified) neuron model that replicates all characteristics virtually impossible (also see[56]). As an alternative approach, Dendrify's ease of implementation and simulation efficiency allows for the development of multiple, different single-neuronal models, each designed to replicate specific features found in these cells.

The dendrites of biological CA1 PCs express several VGICs that allow them to perform complex operations[10,11,16]. For simplicity, we equipped our CA1 neuron model only with Na⁺ VGICs, which underlie the generation of Na⁺ dSpikes (Fig. 3). First, to test our model's ability

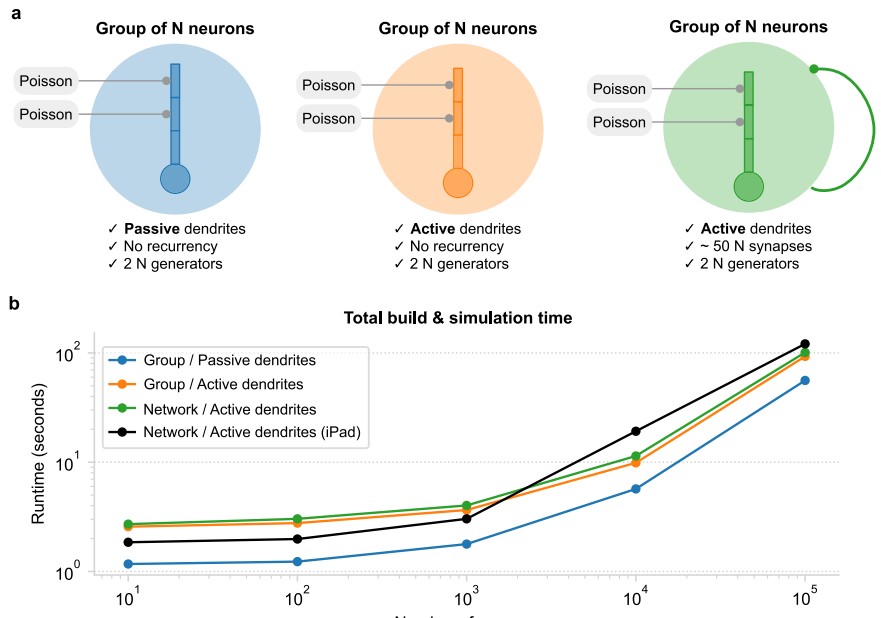

**Fig. 6 | Estimating Dendrify's performance for increasing network complexity and size. a** Schematic illustration of the three model cases used for the scalability analysis. In all cases, the neuronal model was an adapted version of the four-compartment model shown in Fig. 2a. Note that the number of Poisson input generators scaled with *N*. Left: a group of *N* neurons with passive dendrites and no recurrent synapses. Middle: a group of *N* neurons with active dendrites (i.e., furnished with Na+ dSpikes) and no recurrent synapses. Right: a recurrent network of *N* neurons with active dendrites and ~50 synapses/neuron. **b** Scalability plots, showing how the combined build and simulation time scales when increasing *N*. The times plotted here represent the average of 10 runs. Simulations were performed on a laptop (blue, orange, and green) or an iPad (black). For more information, refer to Supplementary Table 4. All scalability codes and the raw results are available on GitHub.

to generate BPAPs, we injected current at the soma (500 ms square pulse of 200 pA) and recorded simultaneously from the most distal parts of the apical dendritic segments (Fig. 4f). We observed that BPAPs are successfully generated and propagate robustly to the end of the main apical trunk (250 μm from the soma). From that point onwards (>250 μm from the soma), BPAPs are reduced to small-amplitude spikelets that fail to trigger dSpike initiation in the distal dendritic segments. This phenomenon has also been documented in recent in vitro studies[53]. However, we should note that back-propagation efficacy among actual CA1 PCs is quite variable and highly dependent on the dendritic morphology and ionic channel distribution[68].

Next, we tested our model's ability to generate dSpikes in response to correlated synaptic input onto its oblique dendrites (see Supplementary Fig. 5). This property is a hallmark of real CA1 PCs[48] and has been used as a metric of model accuracy[56]. Our model reproduces a sigmoid-like input–output function (Fig. 4g), also observed in a previous example (Fig. 3f). Above a certain number of quasi-simultaneous activation (0.1 ms intervals) of synaptic inputs, dendritic responses increase sharply due to dSpike initiation, resulting in supralinear integration[44]. Dendritic sodium spikes cause a rapid jump in the amplitude and kinetics of somatic EPSPs, similar to what is observed in in vitro and biophysical modeling studies[48,67] (Fig. 4h). Capturing this dendro-somatic nonlinear interaction in our model is essential since this feature is known to increase the conditional excitability of biological CA1 PCs and the temporal precision of their spiking output[11,17].

In sum, the above example demonstrates that Dendrify can be used to build versatile, reduced models that reproduce a wide range of biophysical and synaptic characteristics of specific neuronal types. Although at a fraction of the computational cost, these reduced models are on par with far more complex ones in terms of accuracy for several features. Moreover, their small number of parameters makes them substantially more flexible and tractable since modelers can easily adjust their properties and incorporate any available data type.

## Example 4: Pathway interaction in CA1 model neurons

Biological CA1 PCs are more likely to generate action potentials when input from the EC on their distal tuft is paired with coincident CA3 input on more proximal dendritic branches. Due to strong dendritic attenuation, distal synaptic input generally has a negligible effect on the soma, even when dSpikes are generated[69]. However, combining EC and (moderate) CA3 input results in more reliable dSpike initiation and propagation, facilitating axonal action-potential output[69].

To test whether our reduced model (Fig. 4a) captures the coincidence detection capabilities of CA1 pyramidal neurons, we constructed a pool of 10,000 CA1 pyramidal neurons (Fig. 5a). Every neuron received five streams of input drawn from two different Poisson distributions (EC vs. CA3). Each input stream was assigned to a single dendritic branch; two EC streams impinged onto the distal tuft segments, whereas three CA3 streams impinged onto the oblique dendrites and the distal trunk. To replicate the experiments of Jarsky et al.[69] regarding the response of CA1 pyramidal neurons to EC, CA3, and EC + CA3 input, we adjusted the average rates (*λ*) of the Poisson distributions so that: (a) When only the EC pathway is active, neurons have a moderate probability (>55%) of generating at least one distal dSpike, but no somatic APs (Supplementary Figs. 6a and 7a). (b) When only the CA3 pathway is active, neurons generate neither dendritic nor somatic spikes (Supplementary Figs. 6b and 7b). (c) The model output when simultaneously activating the two input pathways in the presence or absence of dendritic Na+ VGICs is shown in (Fig. 5b, c, Supplementary Fig. 7c, d).

In control conditions (dSpikes ON), most neurons (~80%) generated one or more somatic spikes when both the EC and CA3 pathways were active. The rest of the population remained silent throughout the 500 ms of the simulation duration. Deactivating dendritic spikes (dSpikes OFF) impacted neuronal firing significantly: the percentage of active neurons dropped to ~10%, signifying a ~70% decrease compared to the control experiment (dSpikes ON). In addition, all active neurons fired just a single somatic spike. This finding is in line with previous

studies[69] and suggests a direct link between dendritic spiking and the conditional activation of CA1 PCs. Importantly, it highlights our model's ability to reproduce complex pathway interaction rules discovered in biological neurons beyond their basic dendritic properties (Fig. 4).

We next performed a parametric exploration of the input space to gain more insight into the above phenomenon and assess its robustness (Fig. 5d). Specifically, we created ten input distributions for each pathway, with firing rates that varied by 50–150% (with step 10%) of the original values. This led to 121 EC/CA3 input combinations, which were tested in the presence and absence of dSpikes. Coincidence detection efficacy was estimated using the mean neuronal firing rate (MFR) for every combination of inputs (Fig. 5d left, center). This metric provides a quantitive way of gauging the dendritic effect on somatic output (Fig. 5b) rather than simply recording the percentage of active neurons.

We found that dSpike deactivation greatly decreased the estimated MFR across all input combinations (Fig. 5d right). This drop in MFR ranged between 40 and 100% (Fig. 5e); cases with lower initial activity were prone to complete silencing, whereas high-activity cases were affected to a lesser extent. Moreover, dendritic spiking significantly decreased the inter-spike intervals (ISI) of somatic APs (Fig. 5f). The increased excitability caused by dSpikes resulted in somatic responses with lower ISIs, close to those reported during bursting. However, in agreement with experimental data[70,71], the simulated neurons did not generate actual somatic bursts since this behavior requires the presence of dendritic $Ca^{2+}$ plateau potentials, which are not included in this model.

Overall, this example highlighted the ability of our simplified neuron models to reproduce coincidence detection rules intrinsic to the dendrites of biological CA1 PCs. Moreover, we verified the robustness of this behavior through a wide variety of EC/CA3 input parameters. Finally, we showed that dendritic $Na^+$ spikes determine the frequency of somatic output in response to coincident input and their temporal precision, reducing the threshold for strong somatic activity[70].

### Scalability analysis

We have shown that reduced compartmental I&F models, equipped with active, event-driven dendritic mechanisms, can reproduce numerous realistic dendritic functions. However, point-neuron models are currently the gold standard for SNN research thanks to their simplicity, efficiency, and scalability. To assess the viability of our approach for large-network simulations, we tested how Dendrify's performance scales with increasing network size and complexity (Fig. 6). It is important to note that since simulation performance depends on multiple factors such as model complexity, hardware specifications, and case-specific optimizations (e.g., C++ code generation[42] or GPU acceleration[72,73]), designing a single most representative test is unrealistic. For the sake of simplicity and to replicate a real-world usage scenario, all simulations presented in this section were performed on an average laptop using standard and widely used Python tools (Supplementary Table 4). We also run the most demanding test case on an iPad to showcase our approach's universal compatibility and low computational cost.

For the simplest test case (Fig. 6a left), we simulated a group of four-compartment neurons with passive dendrites (adapted from Fig. 2). Each neuron received AMPA-like synaptic inputs (Supplementary Table 4) to its distal and medial compartments from two independent Poisson generators. Apart from the external sources of input (two synapses/neurons), no other synaptic connections were included. For the second test (Fig. 6a middle), we created a model identical to the one described above, but instead of having passive dendrites, it was equipped with $Na^+$-type VGICs. For the last test (Fig. 6a right), we created a recurrent network of N neurons with active dendrites. The connection probability was adjusted for each N to ensure that neurons received a fixed number of synapses (~50 synapses/neuron), regardless of network size.

As expected, simulation times increase as a function of N regardless of model complexity (Fig. 6b). Moreover, introducing more mechanisms, such as active dendritic channels or recurrent synapses, impacted performance significantly. However, simulation times remained within reasonable margins for all model cases and increased in a similar manner as N increased. When $N \leq 10^3$, building a model and running a 1-s-long simulation required no more than 4 s to complete, even in the case of the recurrent network. For the same test, increasing N to $10^4$ or $10^5$ resulted in a total runtime of ~11 and ~101 s, respectively. Surprisingly, when running the same test on an iPad, the code not only ran without any modifications but also faster than the Linux laptop for $N < 10^4$.

Overall, these comparisons suggest that Dendrify is a good alternative for developing SNNs that account for various biological features. It combines flexibility with ease of implementation while offering increased exploratory power at a reasonable computational cost. Moreover, its universal compatibility, efficiency, and flexibility make it a great tool for both research and educational purposes.

## Discussion

Establishing a rapport between biological and artificial neural networks is necessary for understanding and hopefully replicating our brain's superior computing capabilities[4,5,74]. However, despite decades of research revealing the central role of dendrites in neuronal information processing[10,11,16,43], the dendritic contributions to network-level functions remain largely unexplored. Dendrify aims to promote the development of realistic spiking network models by providing a theoretical framework and a modeling toolkit for efficiently adding bioinspired dendritic mechanisms to SNNs. This is materialized by developing simplified yet biologically accurate neuron models optimal for network simulations in the Brian 2 simulator[42].

Here, we demonstrated the ability of simple phenomenological models developed with Dendrify to reproduce numerous experimentally observed dendritic functions. First, we showed that even a generic toy model with passive dendrites can display some electrical segmentation due to the resistance between its compartments (Fig. 2). This property allows dendrites to operate semi-autonomously from the soma and multiple input integration sites to coexist within a single neuron[44]. Next, we showed that adding dendritic $Na^+$ VGICs to a basic four-compartment model (Fig. 3) unlocks important dendritic features that include: (a) the presence of branch-specific integration rules affected by local dendritic morphology[43], (b) the supralinear summation of correlated synaptic inputs and its impact on neuronal output[44], (c) the generation of BPAPs as feedback signals of neuronal activity[45,52,68]. Finally, we built a simplified yet biologically constrained model of a CA1 PC (Fig. 4) and showed its ability to capture numerous passive ($\tau_m$, $R_{input}$, sag ratio, somatodendritic attenuation) and active (F–I curve, nonlinear dendritic integration, BPAPs generation) properties of real CA1 PCs. Notably, the reduced model reproduced complex coincidence detection rules found in CA1 PC dendrites and the impact of $Na^+$ dSpikes on the frequency and the temporal precision of neuronal output[17,75] (Fig. 5). Importantly, our scalability tests showed that Dendrify allows the simulation of both single neurons and networks of increasing complexity with a relatively low computational cost, thus making it an ideal tool for the development of bioinspired SNNs. Overall, we illustrated that Dendrify allows for building simple, mathematically tractable models replicating essential dendritic functions and their influence on neuronal activity.

Multiple recent SNNs studies seemingly converge to the same conclusion; neural heterogeneity within a network can positively impact its learning and information processing capabilities[5]. For example, heterogeneous SNNs with dynamic neuronal properties, such as learnable adaptation[76] and membrane time constants[77] or a

slowly moving firing threshold[78], performed better in complex tasks like image classification or playing Atari games. Since dendrites constitute a significant source of heterogeneity in biological networks, we expect that transferring their properties into SNNs can confer important computational advantages. These include (a) the coexistence of numerous semi-independent integration sites within a single neuron[43], (b) flexible and adaptive information processing that adjusts to computational demand[79], (c) the presence of multi-timescale dynamics[46], and (d) synergy between different synaptic plasticity rules[27]. Indeed, few recent studies suggest that combining nonlinear dendritic mechanisms with local learning rules gives SNNs compelling advantages over previous modeling standards. In particular, dendritic SNNs prolong memory retention in an associative task[27], allow the storage of memories using fewer resources[13], and enable sophisticated credit assignment in hierarchical circuits[80]. However, despite noteworthy progress, we have a long way to go until we fully understand the implications of dendritic processing in neural network functions.

Several tools for simulating multicompartmental neurons and networks of such neurons have been developed throughout the years[81]. Among them, the most popular and widely used are NEURON[82], GENESIS[83], NEST[84], and Brian 2[42], and, more recently, Arbor[85]. However, most of these tools use the HH formalism to simulate the active properties of dendrites, either directly or via its implementation in a secondary, low-level programming language, such as NMODL[86] (NEURON and Arbor) of NESTML[87] (NEST). While users can incorporate new features and mechanisms in models developed with these tools, flexibility and ease of implementation can better be achieved by using a single, general-purpose programming language like Python. Moreover, Brian 2 offers a multicompartment neuronal model in its library, with all the advantages of the simulator. However, there is currently no straightforward way to implement a network of such multicompartmental models in Brian 2.

Dendrify is not another simulator like the ones mentioned above. Instead, Dendrify capitalizes on the intuitiveness and powerful features of the Brian 2 simulator, which requires only basic knowledge of the Python programming language. Its aspiration is to facilitate the development of reduced phenomenological neuron models that preserve many essential properties of their biological counterparts in an efficient and flexible manner. It is designed for non-experts to increase its attractiveness to both experimental and theoretical groups interested in developing bioinspired SNNs. Instead of relying on the HH formalism to simulate VGICs[33,57], dSpike mechanisms are modeled in an event-driven fashion, thus significantly reducing model complexity while maintaining high biological accuracy. Moreover, contrary to similar known approaches[27], in Dendrify, the dSpikes and BPAPs are not simulated by clamping segment voltages. Thus, our implementation allows multiple synaptic or dendritic currents to be summed as in real neurons. Notably, the proposed approach requires a relatively small number of free parameters, resulting in straightforward model development and calibration. Another advantage of our implementation is its compatibility with all popular operating systems running on CPUs and GPUs[72,73]. Finally, our approach allows testing new algorithms compatible with neuromorphic hardware[88–90], which has seen impressive resource-saving benefits by including dendrites[91]. We expect Dendrify to be a valuable tool for anyone interested in developing SNNs with a high degree of bioinspiration to study how single-cell properties can influence network-level functions.

It is important to note that the presented modeling framework does not come without limitations. First, reduced compartmental models cannot compete with morphologically detailed models in terms of spatial resolution. More specifically, in neuronal models with detailed morphologies, each dendritic section consists of several segments used to ensure numerical simulation stability and allow more sophisticated and realistic synaptic placement. By contrast, with Dendrify, we aim to simply extend the point-neuron model by adding a few compartments that account for specific regions in the dendritic morphology. Another limitation is that Dendrify currently depends on Brian's explicit integration methods to solve the equations of the reduced compartmental models. While this approach improves performance, it limits the number of compartments that can be simulated without loss of numerical accuracy[92]. Since Dendrify is commonly used for neuron models with a small number of big compartments, we expect that explicit approaches and a reasonable simulation time step would not cause any substantial numerical issues. To test this, we directly compared Dendrify against SpatialNeuron (which utilizes an implicit method) using an adapted version of the four-compartment model shown in Fig. 3. We show a model with few dendritic compartments and a relatively small integration time step ($dt \leq 0.1$ ms), results in almost identical responses to Brian's SpatialNeuron (Supplementary Figs. 8–17).

Another limitation pertains to our event-based implementation of spikes. Since we do not utilize the HH formalism, certain experimentally observed phenomena cannot be replicated by the standard models provided with Dendrify. These include the depolarization block emerging in response to strong current injections[93] or the reduction of backpropagation efficiency observed in some neuronal types during prolonged somatic activity[68]. Moreover, the current version of Dendrify supports only Na$^+$ and partially Ca$^{2+}$ VGICs and ignores another known ion channel types[94]. Finally, synaptic plasticity rules must be manually implemented using standard Brian 2 objects. However, Dendrify is a project in continuous development, and based on the community feedback, many new features or improvements will be included in future updates.

In summary, we introduced a theoretical framework and a set of tools to allow the seamless development of reduced yet realistic spiking models of any neuronal type. We hope the tool will be readily adopted by neuroscientists and neuromorphic engineers, facilitating knowledge discovery while advancing the development of powerful brain-inspired artificial computing systems.

## Methods
### Somatic compartment
The CA1 PC neuronal model is simulated as a leaky I&F model (Eq. 1) with conductance-based adaptation (Eq. 2).

$$C_m^s \frac{dV_m^s}{dt} = -\bar{g}_L^s (V_m^s - E_L^s) - g_A (V_m^s - E_A) + \sum_{i \in \mathcal{C}^s} I_a^{i,s} + \sum_{j \in \mathcal{S}^s} I_{syn}^{j,s} + I_{ext}^s \quad (1)$$

$$\tau_A \frac{dg_A}{dt} = \bar{g}_A |V_m^s - V_A| - g_A \quad (2)$$

where $V_m^s$ denotes the somatic membrane voltage, $C_m^s$ the membrane capacitance, $\bar{g}_L^s$ the constant leak conductance, $E_L^s$ the leak reversal potential, $g_A$ the adaptation conductance, $E_A$ the adaptation reversal potential, $I_a^{i,s}$ the axial current from the $i$th compartment connected to the soma, $\mathcal{C}^s$ the set with all compartments that are connected with the somatic compartment, $I_{syn}^{j,s}$ a current describing the effect of synaptic input from the $j$th presynaptic neuron to the soma, $\mathcal{S}^s$ a set with the presynaptic neurons connected to the soma, and $I_{ext}^s$ denotes an external current injected into the somatic compartment (similar to an intracellular electrode). The adaptive conductance is changing over time, with $\tau_A$ denoting the time constant of the adaptation and $\bar{g}_A$ is the maximum conductance of the adaptation current. $|\cdot|$ denotes the absolute value.

When the somatic voltage crosses a threshold, $V_{th}$, a spike is generated. Here, we modified the traditional approach of the I&F models, where after a spike generation, the voltage resets back to a

predetermined value, $V_{\text{reset}}$, and we include two resets, one that drives the voltage instantly to a high value, $V_{\text{spike}}$, to account for the biological spike amplitude, and we incrementally increase the $g_A$ by a constant amount $b$, to account for the spike-triggered adaptation, and then after a short decay, we instantly reset the voltage to $V_{\text{reset}}$. Mathematically, we describe this process as

$$\text{if } V_m^s > V_{th} \text{ then} \begin{cases} V_m^s \leftarrow V_{\text{spike}} \\ g_A \leftarrow g_A + b \\ t_{\text{spike}} \leftarrow t \end{cases}$$

$$\text{if } t = t_{\text{spike}} + 0.5 \, \text{ms then } V_m^s \leftarrow V_{\text{reset}}$$

## Dendritic compartments

The dendritic compartments are governed by a similar equation to the somatic one (Eq. 3) for their dynamics, without the adaptation current and by adding two additional currents that control the dynamics of the dendritic spikes (Eqs. 4–7).

$$C_m^d \frac{dV_m^d}{dt} = -\bar{g}_L^d \left( V_m^d - E_L^d \right) + \sum_{i \in C^d} I_a^{i,d} + \sum_{j \in S^d} I_{\text{syn}}^{j,d} + I_{\text{Na}}^d + I_{K_{\text{dr}}}^d + I_{\text{ext}}^d \quad (3)$$

$$I_{\text{Na}}^d = -g_{\text{Na}}^d \left( V_m^d - E_{\text{Na}} \right) f_{\text{Na}} \quad (4)$$

$$I_{K_{\text{dr}}}^d = -g_{K_{\text{dr}}}^d \left( V_m^d - E_K \right) f_{K_{\text{dr}}} \quad (5)$$

$$\tau_{\text{Na}} \frac{dI_{\text{Na}}^d}{dt} = -I_{\text{Na}}^d \quad (6)$$

$$\tau_{K_{\text{dr}}} \frac{dI_{K_{\text{dr}}}^d}{dt} = -I_{K_{\text{dr}}}^d \quad (7)$$

where the $I_{\text{Na}}^d$ and $I_{K_{\text{dr}}}^d$ denote the sodium (Na$^+$) and the delayed-rectified potassium (K$^+$) currents, respectively. $g_{\text{Na}}^d$ and $g_{K_{\text{dr}}}^d$ are the corresponding conductances. These currents are simulated as exponential decay, with time constants $\tau_{\text{Na}}$ and $\tau_{K_{\text{dr}}}$, respectively. $f_{\text{Na}}$ and $f_{K_{\text{dr}}}$ are Boolean parameters indicating the generation of a dendritic spike.

## Dendritic spike mechanism

To activate the sodium current, the $V_m^d$ must cross a threshold, $f_{\text{Na}}$ to be equal to 1, and to be outside of the refractory period of the sodium current:

$$\text{if} \begin{cases} V_m^d > V_{th}^d \\ f_{\text{Na}} = 1 \\ t > t_{\text{spike}}^d + t_{\text{ref}}^{\text{Na}} \end{cases} \text{then} \begin{cases} g_{\text{Na}}^d \leftarrow g_{\text{Na}}^d + \bar{g}_{\text{Na}}^d \\ f_{\text{Na}} \leftarrow 0 \\ f_{K_{\text{dr}}} \leftarrow 1 \\ t_{\text{spike}}^d \leftarrow t \end{cases}$$

where $t_{\text{ref}}^{\text{Na}}$ is the refractory period during which another dendritic spike cannot be generated, $\bar{g}_{\text{Na}}^d$ is the increase in conductance, and $t_{\text{spike}}^d$ denotes the time that voltage crosses the threshold.

To activate the potassium current, a time delay should have passed and $f_{K_{\text{dr}}}$ should be equal to 1.

$$\text{if} \begin{cases} t > t_{\text{spike}}^d + t_{\text{offset}}^{K_{\text{dr}}} \\ f_{K_{\text{dr}}} = 1 \end{cases} \text{then} \begin{cases} g_{K_{\text{dr}}}^d \leftarrow g_{K_{\text{dr}}}^d + \bar{g}_{K_{\text{dr}}}^d \\ f_{\text{Na}} \leftarrow 1 \\ f_{K_{\text{dr}}} \leftarrow 0 \end{cases}$$

where $t_{\text{offset}}^{K_{\text{dr}}}$ denotes the time delay in potassium current generation and $\bar{g}_{K_{\text{dr}}}^d$ is the increase in conductance.

In particular, when the dendritic membrane voltage crosses a threshold, a sodium current is applied, and after a delayed time, a potassium current is generated.

## Axial currents between compartments

Each compartment receives an axial current as a sum of all axial currents flowing toward it and coming from the connected compartments (Eq. 8).

$$I_a^k = \sum_{i \in C^k} I_a^{i,k} \quad (8)$$

where $C^k$ denotes all compartments that are connected with the $k$th compartment. Each compartment-specific axial current (Eq. 9) is given by

$$I_a^{i,k} = g_c^{i,k} \left( V_m^k - V_m^i \right) \quad (9)$$

where the $g_c^{i,k}$ denotes the coupling conductance between the $i$th and $k$th compartments.

We use two approaches to calculate the $g_c^{i,k}$ based on the morphological properties of the compartments.

When the total number of compartments is low and the adjacent-to-soma compartments are highly coupled with the soma, we calculate the absolute longitudinal resistance, $R_{\text{long}}$, in $\Omega$ (Eq. 10).

$$R_{\text{long}} = \frac{r_a l^k}{\pi \left( \frac{d^k}{2} \right)^2} \quad (10)$$

Thus, the coupling conductance is, by definition, the reverse of $R_{\text{long}}$ (Eq. 11).

$$g_c^{i,k} = \frac{1}{R_{\text{long}}} \quad (11)$$

where $d^k$ denotes the diameter of the $k$th compartment, $l^k$ its length, and $r_a$ its specific axial resistance in $\Omega$ cm. The coupling conductance is given in S (siemens). Thus, the axial current is calculated in absolute units, i.e., A (ampere).

The second method uses the half-cylinder approach, where the coupling term of two adjacent compartments (e.g., $k$th and $i$th compartment, respectively) is calculated between their centers (Eq. 12). In this case, $R_{\text{long}}$ is given by:

$$R_{\text{long}} = \frac{1}{2} \left( \frac{r_a l^k}{\pi \left( \frac{d^k}{2} \right)^2} + \frac{r_a l^i}{\pi \left( \frac{d^i}{2} \right)^2} \right) \quad (12)$$

Again, the coupling conductance is calculated as the inverse of $R_{\text{long}}$ (Eq. 13).

$$g_c^{i,k} = \frac{1}{R_{\text{long}}} \quad (13)$$

Notice that we did not divide by the surface area of interest as we wrote the differential equations in absolute terms. Thus, two adjacent compartments have the same coupling conductance $g_c^{i,k} = g_c^{k,i}$.

## Global and specific properties

We assume that all compartments are cylinders with known diameter $d$ and length $l$. The surface area of the $i$th compartment is calculated

using the open cylinder geometry (Eq. 14).

$$A^i = 2\pi \left(\frac{d^i}{2}\right) t^i \qquad (14)$$

and its total membrane capacitance (Eq. 15) and leak conductance (Eq. 16) are given by:

$$C_m^i = c_m^i A^i \qquad (15)$$

$$\bar{g}_L^i = \frac{1}{r_m^i} A^i \qquad (16)$$

where $c_m^i$ is the specific capacitance in $\mu F \cdot cm^{-2}$ and $r_m^i$ is the specific membrane resistivity in $\Omega \cdot cm^2$.

## Synaptic currents

The synaptic currents that can flow to each compartment can be AMPA, NMDA, or GABA (Eq. 17). The mathematical description is:

$$I_{\text{syn}}^i(t) = \bar{g}_{\text{syn}}^i f_{\text{syn}}\left(\tau_{\text{syn}}^{\text{rise}}, \tau_{\text{syn}}^{\text{decay}}\right) s_{\text{syn}}^i(t)\left(V_m^i - E_{\text{syn}}\right)\sigma\left(V_m^i\right) \qquad (17)$$

where $f_{\text{syn}}(\tau_{\text{syn}}^{\text{rise}}, \tau_{\text{syn}}^{\text{decay}})$ is a normalization factor dependent on the rise and decay time constants ($\tau_{\text{syn}}^{\text{rise}}$ and $\tau_{\text{syn}}^{\text{decay}}$) to ensure that for every presynaptic spike, the maximum conductance is $\bar{g}_{\text{syn}}^i$, i.e., the $f_{\text{syn}}(\tau_{\text{syn}}^{\text{rise}}, \tau_{\text{syn}}^{\text{decay}})s_{\text{syn}}^i(t)$ term is bounded in [0,1]. Subscript *syn* denotes the type of synapse, i.e., syn $\in$ {AMPA,*NMDA*,*GABA*}.

The $s_{syn}^i(t)$ term denotes the time dependence of the synaptic conductance. Here, we use two methods; one with a dual exponential form (Eq. 18), as we want to set the rise and decay times independently, and the other as a simple exponential decay (Eq. 19).

$$s_{\text{syn}}^i(t) = H\left(t - t_{\text{pre}}\right)\left(\exp\left(-\frac{t - t_{\text{pre}}}{\tau_{\text{syn}}^{\text{decay}}}\right) - \exp\left(-\frac{t - t_{\text{pre}}}{\tau_{\text{syn}}^{\text{rise}}}\right)\right) \qquad (18)$$

$$s_{\text{syn}}^i(t) = H\left(t - t_{\text{pre}}\right)\exp\left(-\frac{t - t_{\text{pre}}}{\tau_{\text{syn}}^{\text{decay}}}\right) \qquad (19)$$

where H(·) denotes the Heaviside function (Eq. 20).

$$H(z) = \begin{cases} 1, \text{if } z \geq 0 \\ 0, \text{if } z < 0 \end{cases} \qquad (20)$$

The normalization factor is the peak value of $s_{\text{syn}}^i$ at time $t_{\text{peak}}$, and we calculate it via zeroed out the derivative of $s_{\text{syn}}^i$ with respect to time $t$ (Eqs. 21–22).

$$\frac{ds_{\text{syn}}^i}{dt} \overset{\text{set}}{=} 0 \Longleftrightarrow t_{\text{peak}} = \frac{\tau_{\text{syn}}^{\text{decay}}\tau_{\text{syn}}^{\text{rise}}}{\tau_{\text{syn}}^{\text{decay}} - \tau_{\text{syn}}^{\text{rise}}} \ln\left(\frac{\tau_{\text{syn}}^{\text{decay}}}{\tau_{\text{syn}}^{\text{rise}}}\right) + t_{\text{pre}} \qquad (21)$$

$$f_{\text{syn}}\left(\tau_{\text{syn}}^{\text{rise}}, \tau_{\text{syn}}^{\text{decay}}\right) = \frac{1}{s_{\text{syn}}^i\left(t_{\text{peak}}\right)} \qquad (22)$$

For AMPA and GABA currents, the voltage dependence is neglected, i.e., $\sigma\left(V_m^i\right) = 1$. For the NMDA currents, which are voltage-dependent due to magnesium ($Mg^{2+}$) blockade (Eq. 23).

$$\sigma\left(V_m^i\right) = \frac{1}{1 + \frac{[Mg^{2+}]_o}{\beta} \cdot \exp\left(-\alpha\left(V_m^i - \gamma\right)\right)} \qquad (23)$$

where $\beta$ (mM), $\alpha$ ($mV^{-1}$), and $\gamma$ (mV) control the magnesium and voltage dependencies, respectively, and $[Mg^{2+}]_o$ denotes the external magnesium concentration, usually set at a predetermined and constant level (in mM).

The dynamics of the synaptic conductance using the dual exponential form (Eq. 18) are given by a set of two differential equations (Eqs. 24–25) that simulate the double exponential relationship found in synapses[95].

$$\frac{ds_{\text{syn}}^i}{dt} = -\frac{s_{\text{syn}}^i}{\tau_{\text{syn}}^{\text{decay}}} + \frac{x_{\text{syn}}^i\left(1 - s_{\text{syn}}^i\right)}{\tau_{\text{syn}}^{\text{rise}}} \qquad (24)$$

$$\frac{dx_{\text{syn}}^i}{dt} = -\frac{x_{\text{syn}}^i}{\tau_{\text{syn}}^{\text{rise}}} \qquad (25)$$

$$\text{if } t = t_{\text{pre}} \text{ then } x_{\text{syn}}^i \leftarrow x_{\text{syn}}^i + 1$$

The dynamics of the synaptic conductance using the single exponential decay form (Eq. 19) are governed by a single differential equation (Eq. 26).

$$\frac{ds_{\text{syn}}^i}{dt} = -\frac{s_{\text{syn}}^i}{\tau_{\text{syn}}^{\text{decay}}} \qquad (26)$$

$$\text{if } t = t_{\text{pre}} \text{ then } s_{\text{syn}}^i \leftarrow s_{\text{syn}}^i + 1.$$

The normalization function when the simple decay method is applied is $f_{\text{syn}} = 1$.

As a compartment can receive more than one presynaptic connection of the same type and/or synapses of different types simultaneously, the total synaptic current of the *i*th compartment is given by the corresponding summation of all incoming currents (Eq. 27).

$$
\begin{aligned}
I_{\text{syn}}^i(t) = {}& \bar{g}_{\text{AMPA}}^i\left(V_m^i - E_{\text{AMPA}}\right)f_{\text{AMPA}}\sum_{j\in S_{\text{AMPA}}^i} s_{\text{AMPA}}^{j,i}(t) \\
& + \bar{g}_{\text{NMDA}}^i\left(V_m^i - E_{\text{NMDA}}\right)f_{\text{NMDA}}\sum_{j\in \mathcal{S}_{\text{NMDA}}^i} s_{\text{NMDA}}^{j,i}(t) \\
& + \bar{g}_{\text{GABA}}^i\left(V_m^i - E_{\text{GABA}}\right)\sigma\left(V_m^i\right)f_{\text{GABA}}\sum_{j\in S_{\text{GABA}}^i} s_{\text{GABA}}^{j,i}(t).
\end{aligned}
\qquad (27)
$$

## Numerical integration

Dendrify is compatible with all explicit integration methods available in Brian 2, e.g., Euler, exponential Euler, Runge–Kutta midpoint, and classical, among others. For more details, see the supplementary material (Supplementary Methods) and the official Brian 2 documentation (brian2.readthedocs.io/en/stable/user/numerical_integration.html).

## A practical guide for developing reduced models with bioinspired properties

Here, we provide a step-by-step guide for developing simplified compartmental models that capture key electrophysiological and anatomical features of their biological counterparts. The proposed protocol relies on the previous work of Bush and Sejnowski[28] and focuses on achieving realistic axial resistance ($r_a$), input resistance ($R_{in}$), and membrane time constant ($\tau_m$), along with accurate positioning of synaptic inputs and ionic conductances. We illustrate this approach by

**a**

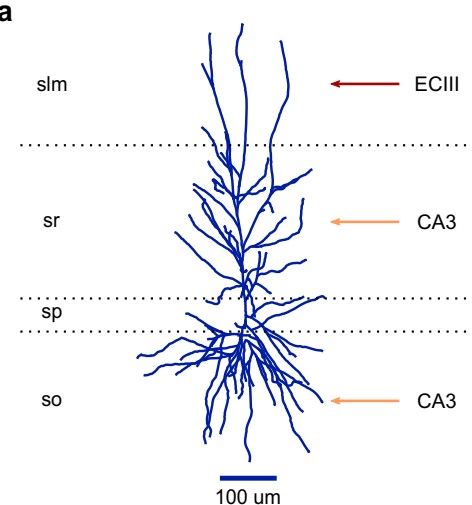

**b**

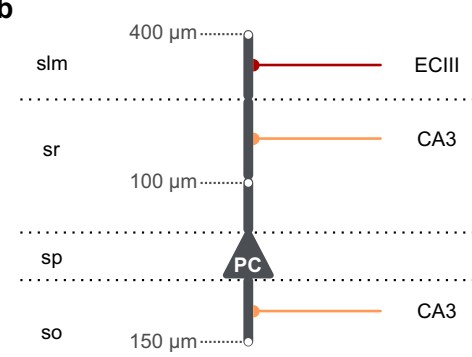

**Fig. 7 | From biological neurons to reduced compartmental neuron models. a** A morphologically detailed reconstruction of a human CA1 PC (adopted from NeuroMorpho.Org[103]). Red arrow: EC layer III input, orange arrows: CA3 input. Horizontal dotted lines: borders of the four CA1 layers (slm: stratum lacunosum-moleculare, sr: stratum radiatum, sp: stratum pyramidale, so: stratum oriens). **b** Schematic illustration of a basic five-compartment CA1 model consisting of a somatic and four dendritic segments (1× basal, 1× proximal trunk, 1× distal trunk, 1× tuft). Grey numbers: distance of the indicated points from the soma. Red axon: EC layer III input, orange axons: CA3 inputs. Horizontal dotted lines: borders of the four CA1 layers as in panel (**a**).

breaking down the development and validation of a reduced CA1 PC (CA1 PC).

Step 1: Identify the most important anatomical and functional regions found in the neuronal morphology (Fig. 7a).

Based on the CA1 region layering and the spatial segregation of external input pathways, CA1 pyramidal neurons can be partitioned into five functionally distinct neuronal regions[96]:

i.   The perisomatic area → primary spiking unit (stratum Pyramidale)
ii.  The basal dendritic area → CA3 input receiver (stratum Oriens)
iii. The proximal apical dendritic area → dendritic region devoid of spines (stratum Radiatum, <100 µm from the soma)
iv.  The medial apical dendritic area → CA3 input receiver (stratum Radiatum, >100 µm from soma)
v.   The distal apical dendritic area → EC layer III input receiver (stratum Lacunosum Moleculare)

Step 2: Design a toy model capturing the main neuronal features identified in the previous step (Fig. 7b).

- Using cylindrical compartments, design a toy model that captures the main morphological features of the neuron of interest. The number of model compartments should not exceed the

number of the identified, functionally unique neuronal regions. This would prevent the model from processing the various input pathways semi-independently, as in real CA1 PCs[69].

- If biological accuracy is more important than simulation performance, the number of compartments can be further increased to account for more neuronal features. For example, adding four compartments to the previous model (see Fig. 4a) allows for accounting for the increased dendritic branching that is observed in the distal, medial and basal areas of CA1 PC dendrites. Other examples of morphologically reduced CA1 models can be seen in ref. [56].

- Set the dimensions of the compartments according to the rules described in ref. [28]. In short, their approach aims to preserve realistic attenuation of the currents traveling along the somatodendritic axis. This is achieved by creating compartments with the correct electrotonic length and a diameter representative of the dendritic diameter observed in real neurons.

- If there is no detailed morphological data, you can set the cylinder lengths that approximate the distance from the soma and capture the decrease in dendritic diameter as we move away from the soma. Due to immense biological variability, the solutions to this problem are infinite, and a single most representative model is impossible to exist.

Step 3: Validation of passive parameters

1) Membrane time constant
   - Start with the values of somatic capacitance ($C_m$) and leakage conductance ($g_L$). Set $C_m$ equal to $1\,\mu F\,cm^{-2}$ and choose the appropriate $g_L$ value so that the desired membrane time constant ($\tau_m$) is achieved according to the formula $\tau_m = C_m/g_L$.
   - Next, use the same values for the dendrites, but we multiply both by a factor of 1.2–2.0 (depending on experimental data, use 1.5 if this value is unknown) to account for the added area due to synaptic spines that are not explicitly modeled.

2) Input resistance and somatodendritic attenuation
   - Set the axial resistance ($R_a$) according to experimental evidence, if available. Typical values range between 100 and 250 MΩ cm.
   - Test the attenuation of currents along the somatodendritic axis by applying long somatic current injections (Fig. 3). By default, Dendrify calculates the coupling conductances according to the half-cylinders formula[97]:

$$R_{\text{long}} = \frac{1}{2}\left(\frac{r_a l^k}{\pi\left(\frac{d^k}{2}\right)^2} + \frac{r_a l^i}{\pi\left(\frac{d^i}{2}\right)^2}\right) \Rightarrow g_c^{i,k} = \frac{1}{R_{\text{long}}} \qquad (28)$$

where superscripts $i$ and $k$ denote two adjacent compartments and $l$, $d$ denote the length and the diameter of the compartments, respectively.

Importantly, small manual corrections might be necessary to achieve more realistic attenuation.

   - Calculate the model's input resistance ($R_{in}$) by using a typical, hyperpolarizing current step protocol[54]. Most likely, the initial values will deviate from the experimental values due to the reduced membrane area of the simplified model. This is why we multiply both $C_m$ and $g_L$ (somatic and dendritic) with the same scale factor until the model reaches the desired $R_{in}$ as explained here[28].

Step 4: Validation of active properties

This step assumes that an I&F model with adaptation is used for the soma, such as the AdEx[98], CAdEx[99], or Izhikevich[100] model. Use somatic current injections to validate the Rheobase and FI curve by adjusting the model variables based on the model-specific guidelines.

Step 5: Validation of dendritic integration

**Article**

The last step includes the validation of the Na$^+$ dendritic spike. First, we set a realistic $g_{Na}$ to $g_K$ ratio, based on experimental evidence. Then, we set a voltage threshold, which denotes the membrane voltage values above which a dSpike is initiated. To account for the geometrical characteristics of the dendritic compartments, we multiply both conductances with the compartmental surface area, i.e., $A^i$. Using the validation protocol depicted in Supplementary Fig. 5, we scale the conductances to capture a realistic dSpike amplitude.

## Data availability

The source code that generates all Figures, as well as the data that support this study, are accessible on GitHub.

## Code availability

The code version underlying this study is available on GitHub and can be accessed on Zenodo[101]. Dendrify can be installed via the Python Package Index (https://pypi.org/project/dendrify/) and is continuously developed and accessible on GitHub. Extensive documentation, including installation instructions, is hosted on https://dendrify.readthedocs.io. Dendrify can be used in Windows, macOS, and Linux operating systems.

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

## Acknowledgements

The authors thank Dr. Marcel Stimberg and Dr. Dan Goodman for their valuable and constructive feedback, Dr. Arjun Masurkar for providing experimental CA1 data, and Dr. Ruben Tikidji-Hamburyan for his input on comparison details among different simulators. The authors also thank the Brian 2 developing team for their continuous support. This work was supported by NIH (1R01MH124867-02) to P.P., the European Commission (H2020-FETOPEN-2018-2019-2020-01, FET-Open Challenging Current Thinking, NEUREKA GA-863245) to P.P., and the Einstein Foundation Berlin, Germany (visiting fellowship, EVF-2019-508) to P.P.

## Author contributions

M.P., S.C., and P.P. conceived the project, designed the experiments, and revised the paper. M.P. developed the Dendrify package, performed the experiments and the benchmark tests, analyzed the data, designed the figures, and wrote the initial draft of the paper. S.C. conceptualized the mathematical implementation of the reduced models, wrote the Methods, and supervised M.P. P.P. supervised and funded the project. All authors contributed to the article and approved the submitted version.

## Competing interests

The authors declare no competing interests.
