## [Peer Review File · Nature Communications]

Introducing the Dendrify framework for incorporating dendrites to spiking neural networksEditorial Note: Parts of this Peer Review File have been redacted as the reviewer did not wish to be named.

REVIEWER COMMENTS

Reviewer #1 (Remarks to the Author):

Dear Authors!

You present the software tool Dendriify. A Python package that seeks to extend Brian2 with primitives to allow for modelling of simplified multi-compartment neuron models. The topic is of interest to both the computational neuroscience and neuromorphic community. The paper is written well, text and figures are clear and engaging.

The manuscript presents Dendriify on four examples of increasing complexity. A basic compartmental model with passive dendrites, a reduced compartmental model with active dendrites, a simplified yet biologically accurate model of a CA1 pyramidal cell, and on pathway interaction in CA1 model neurons. All models come with ready-to-use code examples that reproduce the figures, which is very welcome!

The method section of the paper details the mathematical background of the implementation and gives a guide on developing reduced models.

The manuscript is primarily advertised as presenting a software tool, therefore I will also mainly judge it under this premise.

Please find points that need to be addressed below.

A) Why a new tool?

"To address the abovementioned complexity issues and provide a framework that allows the seamless incorporation of dendrites in SNN models, we developed Dendriify."

Developing a new tool is always fun, but you need to explain why the existing solutions are not sufficient. A convincing way to do so would be to compare one of your examples with implementations in other simulators and point out their shortcomings.

The comparison should be against multi-compartment simulators like NEURON (<https://neuron.yale.edu/>) and/or Arbor (<https://arbor-sim.org/>), but especially the built-in SpatialNeuron of Brian2 itself. Also NEST has support for multi-compartment neurons and should be included in the comparison.

B) The solution to the cable equation

Dendriify collects a set of equations from the currents between compartments and the state variables of neurons and synapses. While this is fine, using explicit methods like forward Euler is not. The equation of the cable is stiff and must be solved via implicit methods like backward Euler. Tools like NEURON and Arbor, but also the SpatialNeuron of Brian2 do it correctly. This is a major problem and

needs to be corrected. I would propose to eliminate your manual implementation and base it on Brian2's SpatialNeuron.

Please see Mascagni, Michael V., and Arthur S. Sherman. "Numerical Methods for Neuronal Modeling." *Methods in Neuronal Modeling 2* (1998). https://www.cs.fsu.edu/~mascagni/papers/RCEV1996_1.pdf

I've put together an example demonstrating the problem.

C) Software best practices

The Python code looks reasonable in form and style. However, you need to implement the following:

- * continuous integration and tests (also against upstream Brian2)
- * continuous delivery, e.g. provide the latest release on PyPI
- * linting and style checking
- * published documentation, e.g. readthedocs.org

You and your users want to use the tool for many years. But without the points above, it will degrade. Especially the checks against upstream Brian2 are vital.

D) Performance

Please show the performance of your implementation. How long does it take to simulate one neuron, 10, 1000? What about networks of neurons?

Minor points

1) You stress that Brian2 has seamless compatibility with Dendrifly which is a bit funny as Dendrifly is based on Brian2 and not the other way around.

2) All examples work with single neurons although you often, especially in the introduction, talk about networks. Please add at least one example of a network of multi-compartment neurons.

3) "The attenuation of currents traveling along the somatodendritic axis is an intrinsic property of biological neurons and is due to the morphology and cable properties of dendritic trees."

You don't really show currents traveling along the axis as you have only a handful of compartments.

4) neuromorphic implementations

Please cite more multi-compartment neuromorphic implementations, e.g.

"Emulating Dendritic Computing Paradigms on Analog Neuromorphic Hardware"
Kaiser et al.
<https://doi.org/10.1016/j.neuroscience.2021.08.013>

Thanks,

your reviewer

```

#!/usr/bin/env python3
"""
Example showing the problem using forward Euler for stiff equations

Based on "Numerical Integration in Computational Neuroscience"
https://www.neuron.yale.edu/ftp/ted/neuron/numerical_integration.pdf

[REDACTED]
"""
import matplotlib.pyplot as plt

from dendrify import Soma, Dendrite, NeuronModel

import brian2 as b
from brian2.units import (ms, um, pA, nS, uS, ohm, cm, mV, uF, mvolt,
siemens, volt, Mohm)

soma = Soma('soma', model='passive', length=10*um, diameter=10*um)
spine = Dendrite('spine', model='passive', length=1*um, diameter=1*um)

g = 0.001*siemens/cm**2
C = 1*uF/cm**2
ra = 1.0168*Mohm

# will show non diverging behaviour
# ra *= 10000

edges = [(soma, spine, 1/ra)]

model = NeuronModel(edges,
                    cm=C,
                    gl=g,
                    v_rest=0*mV, # needs v_rest_zero.patch
                    r_axial=None,
                    scale_factor=1,
                    spine_factor=1)

group = b.NeuronGroup(1,
                    model=model.equations,
                    method='euler',
                    namespace=model.parameters)

model.link(group)

group.V_soma = 1*mV
group.V_spine = 0*mV

M = b.StateMonitor(group, ["V_soma", "V_spine"], record=True)
b.defaultclock = 0.1*ms
b.run(1*ms)

plt.plot(M.t/ms, M.V_soma[0]/mV, label="soma")
plt.plot(M.t/ms, M.V_spine[0]/mV, label="spine")

plt.xlim(0, 1)
plt.ylim(0, 1)

```

```
plt.xlabel("t [ms]")  
plt.ylabel("membrane [mV]")  
plt.legend()
```

```
plt.show()
```

```

diff --git a/dendrifly/compartment.py b/dendrifly/compartment.py
old mode 100644
new mode 100755
diff --git a/dendrifly/ephysproperties.py b/dendrifly/ephysproperties.py
index ab707b7..9611726 100644
--- a/dendrifly/ephysproperties.py
+++ b/dendrifly/ephysproperties.py
@@ -93,7 +93,7 @@ class EphysProperties(object):
     d = {}
     error = None

-     if self.v_rest:
+     if self.v_rest is not None:
         d[f"EL_{self.tag}"] = self.v_rest
     else:
         print(f"ERROR: Could not resolve 'EL_{self.tag}'\n")
diff --git a/dendrifly/neuronmodel.py b/dendrifly/neuronmodel.py
index d2baef5..fb20ae7 100644
--- a/dendrifly/neuronmodel.py
+++ b/dendrifly/neuronmodel.py
@@ -146,7 +146,7 @@ class NeuronModel(object):
     i._ephys_object.gl = gl
     if r_axial and (not i._ephys_object.r_axial):
         i._ephys_object.r_axial = r_axial
-     if v_rest and (not i._ephys_object.v_rest):
+     if v_rest is not None and (not i._ephys_object.v_rest):
         i._ephys_object.v_rest = v_rest
     if scale_factor:
         i._ephys_object.scale_factor = scale_factor

```

Reviewer #2 (Remarks to the Author):

The authors introduce a python-package to facilitate the modeling of dendritic compartments in spiking networks. The objective is an easy to use and computationally efficient framework which captures dendritic features currently omitted in the simulation packages.

On the basis of developments in neurophysiology, computational neuroscience, and recent results in the field of machine learning, they outline how accounting for dendritic phenomenon is necessary to facilitate advancements in computational neuroscience and neuromorphic computing.

The authors set up four model scenarios to exemplify dendritic behaviour.

- * Coupling of passive dendrites
- * Compartmental model with active dendrites
- * A compartmental model fit to a CA1 pyramidal cell.
- * Coincidence detection in populations of CA1 model neurons.

The code is very well documented and accessible at their github repository. This will make it very easy to use for anyone with some programming experience.

Lastly, the authors example a practical guide of developing models with their package.

Developing joint, open source software is extremely important to facilitate research progress and to validate results. Here, this package fills an important gap.

We did check the code and did not find technical problems.

The manuscript is well written and well understandable, serving as a good guide for starters.

Comments

The authors mention several times the computational capabilities, and the compatibility of dendrify with brian 2, yet are not clearly mentioning any of the specifics.

Their github actually makes clear that their implementation is fully build on top of the existing brian 2 frameworks. Which is appropriate as everything can be reinterpreted as more complicated neuron models. This should be clarified in the manuscript as well.

On a technical note, the formatting of the manuscript make any mathematical derivations hard to read, and we hope that the next version improves here.

Response to Reviewers

We are excited that both referees find our work interesting and important and are grateful to both of them for their constructive feedback which has been critical for improving our manuscript. We are especially thankful to the first referee whose comments were extremely helpful in filling the gaps in our approach and making our work more attractive/useful to the community. Below, we present analytical answers to all of their comments.

Reviewer #1

Dear Authors!

You present the software tool Dendripy. A Python package that seeks to extend Brian2 with primitives to allow for modeling of simplified multi-compartment neuron models. The topic is of interest to both the computational neuroscience and neuromorphic community. The paper is written well, text and figures are clear and engaging.

The manuscript presents Dendripy on four examples of increasing complexity. A basic compartmental model with passive dendrites, a reduced compartmental model with active dendrites, a simplified yet biologically accurate model of a CA1 pyramidal cell, and on pathway interaction in CA1 model neurons. All models come with ready-to-use code examples that reproduce the figures, which is very welcome!

The method section of the paper details the mathematical background of the implementation and gives a guide on developing reduced models.

The manuscript is primarily advertised as presenting a software tool, therefore I will also mainly judge it under this premise.

Please find points that need to be addressed below.

We thank the reviewer for their positive comments and appreciation of our work.

A) Why a new tool?

“To address the abovementioned complexity issues and provide a framework that allows the seamless incorporation of dendrites in SNN models, we developed Dendripy.”

Developing a new tool is always fun, but you need to explain why the existing solutions are not sufficient. A convincing way to do so would be to compare one of your examples with implementations in other simulators and point out their shortcomings.

The comparison should be against multi-compartment simulators like NEURON (<https://neuron.yale.edu/>) and/or Arbor (<https://arbor-sim.org/>), but especially the built-in SpatialNeuron of Brian2 itself. Also NEST has support for multi-compartment neurons and should be included in the comparison.

The reviewer raises some valid points here since a comparison with other popular simulators was missing from the initial manuscript (now added in the Discussion/Supplement). Before comparing our approach with other tools, we would like to clarify two important things.

A. Dendrify aims to be more than a software tool

The primary goal of this project was to serve as both an inspiration and a means for facilitating the adoption of dendrites in spiking neural networks used for neuroscience research and/or neuromorphic applications. We place special emphasis on facilitating dendritic SNNs as the impact of dendrites at the network level remains poorly understood. We believe that this gap in the literature persists due to the following reasons:

- a. *In vivo* dendritic research remains technically difficult and requires expensive equipment and techniques that few labs can afford.
- b. Detailed biophysical models of neurons with dendrites are too complex and computationally costly for large-network simulations.
- c. SNNs and neuromorphic studies generally prefer simple, “point-neuron” models because they are more efficient, mathematically tractable, and easier to work with.
- d. Many recent findings of dendritic research have not yet found their way into other fields beyond Neuroscience.

Although we cannot do much about point a, in our manuscript, we tried to approach points b-c by:

- I. Presenting the idea of the “extended point neuron” model that requires only a handful of compartments and event-driven mechanisms to simulate many essential dendritic functions.
- II. Highlighting the advantages of reduced compartmental models through carefully designed experiments.
- III. Summarizing significant findings from decades of dendritic research throughout the manuscript, using simple language while having non-expert readers in mind.

B. Dendrify does not aim to “compete” with or substitute other simulators

Dendrify is not a dedicated simulator per se. It is a tool created to perform a specific task; to facilitate the development of SNNs that efficiently simulate essential dendritic mechanisms. While it is compatible with the Brian 2 simulator, it could easily have been based on other simulators. However, the combination of Brian 2 and Dendrify is currently, to our knowledge, the only software solution that combines the following advantages:

- Model definitions and the computational experiment script are written in the same language. This reduces the chance for error and eliminates the need to learn different languages just to run some simulations. It makes computational modeling more accessible for non-experts as well.
- Models are not black boxes; the differential equations are written as Python strings and parameters are Brian 2 units.
- Inexperienced users benefit from our standardized approach for describing models, while advanced users benefit from Brian’s flexibility that allows the definition of arbitrary models.
- Dendritic spiking can be implemented without using the HH equations, using only event-driven mechanisms. This results in models that are more efficient, more tractable and easier to work with.
- Dendrify and Brian run on all operating systems (in numpy mode) and their installation requires only a single line of code.
- Brian offers numerous options for optimization such as C++ generation and GPU acceleration through Brian2Genn and Brian2CUDA.
- Our approach is reasonably fast, offering great explanatory power.

Having said that, in response to the specific comment, we have done the following:

1. Made small changes throughout the manuscript to highlight the positioning of our work with respect to other tools/approaches.
2. Added 2 paragraphs in the Discussion (Pages: 29-30, Lines: 403-422) that refers to other simulators and points out the differences in their approach vs. ours

3. Directly compared our approach to the SpatialNeuron class of Brian 2 (see comment below). We also added an entire section in the supplementary material (see: **Figures S8-S17 | Simulation accuracy and numerical stability analysis**)

B) The solution to the cable equation

Dendrify collects a set of equations from the currents between compartments and the state variables of neurons and synapses. While this is fine, using explicit methods like forward Euler is not. The equation of the cable is stiff and must be solved via implicit methods like backward Euler. Tools like NEURON and Arbor, but also the SpatialNeuron of Brian2 do it correctly. This is a major problem and needs to be corrected. I would propose to eliminate your manual implementation and base it on Brian2's SpatialNeuron.

*Please see Mascagni, Michael V., and Arthur S. Sherman. "Numerical Methods for Neuronal Modeling." *Methods in Neuronal Modeling 2* (1998). https://www.cs.fsu.edu/~mascagni/papers/RCEV1996_1.pdf*

I've put together an example demonstrating the problem.

We thank the reviewer for this excellent remark. Empirically we have seen that using explicit methods is "good enough" for models with a very small number of compartments and short simulation time steps ($dt \leq 0.1$ ms). Notably, Brian 1 natively supports building networks of reduced compartmental neurons, although this feature was dropped in Brian 2. Additionally, our equation-based approach was based on Brian's guidelines:

"Brian 1 offered support for simple multi-compartmental models in the compartments module. This module allowed you to combine the equations for several compartments into a single Equations object. This is only a suitable solution for simple morphologies (e.g. "ball-and-stick" models) but has the advantage over using SpatialNeuron that you can have several of such neurons in a NeuronGroup. If you already have a definition of a model using Brian 1's compartments module, then you can simply print out the equations and use them directly in Brian 2."

However, we should have made Dendrify's limitations (in its current version) clearer to its users and informed them about possible pitfalls related to the problem explained by the reviewer. To address this comment, we took the following actions:

Action 1: We contacted the Brian team to find out if SpatialNeuron can be used as it is for simulating networks of multi-compartmental neurons. Their official response (by Marcel Stimberg, co-creator of Brian) was that: "Using SpatialNeuron is not feasible in your case, since it cannot be used to make networks (there is work in progress to make this possible, but don't expect the feature to land in 2022)."

Conclusion 1: Currently, Brian 2 does not natively support creating neuronal networks using SpatialNeuron objects.

Action 2: We explored the option of basing Dendrify on SpatialNeuron. Although Brian's SpatialNeuron is neither designed nor optimized for network simulations, we could theoretically base Dendrify on it by creating a wrapper "Network" class that uses SpatialNeuron objects at its core. However, this approach would be highly problematic for the reasons we analyze below. Notably, as explained by Marcel Stimberg in a recent CNS tutorial (2:04:00 of this video), Brian simulations run significantly faster and more efficiently when the model code is written in a way that minimizes the usage of Brian objects (NeuronGroups, Synapses, Monitors etc).

Basing our approach on SpatialNeuron would come with the following shortcomings:

- a. Every single node in a network of compartmental neurons should be an additional SpatialNeuron object (i.e., 10^n neurons = 10^n SpatialNeuron objects). Interestingly, under the hood, a SpatialNeuron is, in fact, a NeuronGroup itself. With Dendripy, a single NeuronGroup object is created for every different neuronal population regardless of its size (i.e., a homogenous population of 10^n neurons = 1 NeuronGroup object).
- b. Since every network node would be an independent NeuronGroup, every synapse should be an additional Synapse object due to how synapses are implemented in Brian (i.e., 10^n synapses = 10^n Synapse objects). With Dendripy, each synaptic pathway requires a single Synapse object regardless of the total number of synapses (i.e., one synaptic pathway that results in 10^n synapses = 1 Synapse object).
- c. To record a variable of interest during a simulation (e.g., the voltage of a single compartment or the somatic spike times), one would need to include an extra Monitor object per compartment per neuron. To understand this problem, please consider the following example: Let us assume a Network consisting of 10000, 3-compartmental neurons and that for a given experiment, we wish to record all compartment voltages and the somatic spike times. For a SpatialNeuron-based approach, this experiment would require 30000 voltage Monitors and 10000 spike Monitors. With Dendripy, we can collect the same data using only two monitor objects.

To test how some of the abovementioned implementation differences affect real-world performance, we performed a simple benchmarking test, the details of which are shown below.

Benchmark details

Model:

- 4-compartmental model (3 passive dendrites + leaky I&F somatic unit)
- Input: A square current pulse (1000 ms long) injected at the soma

Simulation details:

- Total simulated time: 1250 ms
- Time step (dt) = 0.1 ms
- Integration method: Forward Euler
- A SpikeMonitor was also used to record somatic spikes

System:

- Ubuntu 22.04.1 laptop
- i7-9750H CPU
- 16 GB of RAM

Setup:

- Jupyter notebook (%%timeit module)
- Run Brian in "numpy" mode
- Measured model build + runtime (mean of 10 runs)

Results (mean execution time \pm std, n=10 runs):

- 1 x Spatial Neuron: **2.18 s \pm 23.9 ms**
- 3 x Spatial Neurons: **5.83 s \pm 101 ms**
- 1 x Dendripy Neuron: **0.985 s \pm 16.5 ms**
- 6000 x Dendripy Neurons: **2.18 s \pm 23 ms**

Even in this trivial example, we observe that Dendripy has significant performance benefits over a SpatialNeuron-based approach. Simulating a single SpatialNeuron required as much time as a group of 6000 neurons made with Dendripy. Furthermore, adding only 2 SpatialNeurons resulted in almost three times slower simulation time. It is

evident that our equation-based approach minimizes the need for creating Brian objects, thus seeming optimal in terms of performance and allowing for a good population size scaling. Notably, we expect the performance gap between the two approaches to be even more apparent if synapses and more monitors are included in the comparison.

Next, we examined how replacing forward Euler with more sophisticated and stable (explicit) integration methods available in Brian 2 would impact performance. The results are shown below:

Exponential Euler (mean execution time \pm std, n=10 runs):

- 1 x Spatial Neuron: **2.14 s \pm 27.2 ms**
- 3 x Spatial Neurons: **5.91 s \pm 195 ms**
- 1 x Dendrify Neuron: **1.06 s \pm 8.42 ms**
- 6000 x Dendrify Neurons: **2.34 s \pm 62.6 ms**

Heun's method (mean execution time \pm std, n=10 runs):

- 1 x Spatial Neuron: **2.19 s \pm 39 ms**
- 3 x Spatial Neurons: **5.98 s \pm 68.8 ms**
- 1 x Dendrify Neuron: **1.01 s \pm 55.2 ms**
- 6000 x Dendrify Neurons: **2.28 s \pm 64.2 ms**

Second Order Runge-Kutta (mean execution time \pm std, n=10 runs):

- 1 x Spatial Neuron: **2.2 s \pm 27.3 ms²**
- 3 x Spatial Neurons: **5.94 s \pm 111 ms**
- 1 x Dendrify Neuron: **1.5 s \pm 12 ms**
- 6000 x Dendrify Neurons: **4.27 s \pm 75 ms**

Fourth Order Runge-Kutta (mean execution time \pm std, n=10 runs):

- 1 x Spatial Neuron: **2.13 s \pm 148 ms**
- 3 x Spatial Neurons: **6.01 s \pm 311 ms**
- 1 x Dendrify Neuron: **2.38 s \pm 31.7 ms per**
- 6000 x Dendrify Neurons: **8.65 s \pm 500 ms**

Comment: even when using more computationally costly integration methods, Dendrify seems to be a more viable solution than a SpatialNeuron-based approach for building large-scale networks. This occurs because simulation time in the case of Dendrify does not increase linearly with the number of neurons.

Apart from the net simulation time, another important aspect of neural network simulations is the amount of memory required to perform an experiment. Less memory-hungry models allow researchers to scale their network's size more easily or increase its complexity and simulation time. For this reason, we also tested the memory cost of a SpatialNeuron vs. an equation-based approach. The model used here is the same as the one described previously. For the memory profiling test, we used the following Python package: memory-profiler · PyPI.

Results summary (memory consumption for creating and running the model):

- 1 x SpatialNeuron: **13.48 MiB or 14.134804 MB** (145.27 MiB \rightarrow 158.75 MiB)
- 1 x Dendrify neuron: **3.09 MiB or 3.2401 MB** (145.31 MiB \rightarrow 148.40 MiB)
- 19000 x Dendrify neurons: **11.98 MiB or 12.56194 MB** (145.55 MiB \rightarrow 157.53 MiB)

Detailed results:

1 x SpatialNeuron

30	145.27 MiB	0.00 MiB	1	morpho = b.Cylinder(length=25*um, diameter=25*um, n=1)
31	145.27 MiB	0.00 MiB	1	morpho.dendrite = b.Cylinder(length=300*um, diameter=1*um, n=3)
32				
33	145.27 MiB	0.00 MiB	1	eqs = '''
34				Im = GL * (EL - v) : amp/meter**2
35				I : amp (point current)
36				'''
37				
38	149.65 MiB	4.38 MiB	2	spatial_neuron = b.SpatialNeuron(morphology=morpho, model=eqs, Cm=CM, Ri=RI,
39	145.27 MiB	0.00 MiB	1	threshold='v > -40*mV', reset='v = -50*mV',
40	145.27 MiB	0.00 MiB	1	refractory=3*ms, threshold_location=0,
41	145.27 MiB	0.00 MiB	1	method=INTEGRATION_METHOD)
42	149.65 MiB	0.00 MiB	1	spatial_neuron.v = EL
43				
44	149.65 MiB	0.00 MiB	1	S = b.SpikeMonitor(spatial_neuron, record=True)
45				
46	158.75 MiB	9.11 MiB	1	b.run(T_START)
47	158.75 MiB	0.00 MiB	1	spatial_neuron.I[0] = I_INJ_SOMA
48	158.75 MiB	0.00 MiB	1	b.run(T_INJ)
49	158.75 MiB	0.00 MiB	1	spatial_neuron.I[0] = 0*amp
50	158.75 MiB	0.00 MiB	1	b.run(T_END)

1 x Dendrify neuron

128	145.31 MiB	0.00 MiB	1	soma = Soma('soma', model='leakyIF', length=25*um, diameter=25*um)
129	145.31 MiB	0.00 MiB	1	dend0 = Dendrite('dend0', length=100*um, diameter=1*um)
130	145.31 MiB	0.00 MiB	1	dend1 = Dendrite('dend1', length=100*um, diameter=1*um)
131	145.31 MiB	0.00 MiB	1	dend2 = Dendrite('dend2', length=100*um, diameter=1*um)
132				
133	145.31 MiB	0.00 MiB	1	connections = [(soma, dend0), (dend0, dend1), (dend1, dend2)]
134	145.31 MiB	0.00 MiB	1	model = NeuronModel(connections, cm=CM, gl=GL, v_rest=EL, r_axial=RI)
135	147.37 MiB	2.06 MiB	2	dend_neuron = b.NeuronGroup(N=1, model=model.equations,
136	145.31 MiB	0.00 MiB	1	method=INTEGRATION_METHOD,
137	145.31 MiB	0.00 MiB	1	threshold='V_soma > -40*mV',
138	145.31 MiB	0.00 MiB	1	reset='V_soma = -50*mV',
139	145.31 MiB	0.00 MiB	1	refractory=3*ms, namespace=model.parameters)
140	147.37 MiB	0.00 MiB	1	dend_neuron.V_soma = EL
141	147.37 MiB	0.00 MiB	1	dend_neuron.V_dend0 = EL
142	147.37 MiB	0.00 MiB	1	dend_neuron.V_dend1 = EL
143	147.37 MiB	0.00 MiB	1	dend_neuron.V_dend2 = EL
144				
145	147.37 MiB	0.00 MiB	1	S = b.SpikeMonitor(dend_neuron, record=True)
146				
147	148.40 MiB	1.03 MiB	1	b.run(T_START)
148	148.40 MiB	0.00 MiB	1	dend_neuron.I_ext_soma = I_INJ_SOMA
149	148.40 MiB	0.00 MiB	1	b.run(T_INJ)
150	148.40 MiB	0.00 MiB	1	dend_neuron.I_ext_soma = 0*amp
151	148.40 MiB	0.00 MiB	1	b.run(T_END)

19000 x Dendrify neurons

128	145.55 MiB	0.00 MiB	1	soma = Soma('soma', model='leakyIF', length=25*um, diameter=25*um)
129	145.55 MiB	0.00 MiB	1	dend0 = Dendrite('dend0', length=100*um, diameter=1*um)
130	145.55 MiB	0.00 MiB	1	dend1 = Dendrite('dend1', length=100*um, diameter=1*um)
131	145.55 MiB	0.00 MiB	1	dend2 = Dendrite('dend2', length=100*um, diameter=1*um)
132				
133	145.55 MiB	0.00 MiB	1	connections = [(soma, dend0), (dend0, dend1), (dend1, dend2)]
134	145.55 MiB	0.00 MiB	1	model = NeuronModel(connections, cm=CM, gl=GL, v_rest=EL, r_axial=RI)
135	148.90 MiB	3.35 MiB	2	dend_neuron = b.NeuronGroup(N=19000, model=model.equations,
136	145.55 MiB	0.00 MiB	1	method=INTEGRATION_METHOD,
137	145.55 MiB	0.00 MiB	1	threshold='V_soma > -40*mV',
138	145.55 MiB	0.00 MiB	1	reset='V_soma = -50*mV',
139	145.55 MiB	0.00 MiB	1	refractory=3*ms, namespace=model.parameters)
140	148.90 MiB	0.00 MiB	1	dend_neuron.V_soma = EL
141	148.90 MiB	0.00 MiB	1	dend_neuron.V_dend0 = EL
142	148.90 MiB	0.00 MiB	1	dend_neuron.V_dend1 = EL
143	148.90 MiB	0.00 MiB	1	dend_neuron.V_dend2 = EL
144				
145	149.16 MiB	0.26 MiB	1	S = b.SpikeMonitor(dend_neuron, record=True)
146				
147	151.77 MiB	2.61 MiB	1	b.run(T_START)
148	151.77 MiB	0.00 MiB	1	dend_neuron.I_ext_soma = I_INJ_SOMA
149	157.53 MiB	5.76 MiB	1	b.run(T_INJ)
150	157.53 MiB	0.00 MiB	1	dend_neuron.I_ext_soma = 0*amp
151	157.53 MiB	0.00 MiB	1	b.run(T_END)

For this specific test, the amount of memory needed to create and simulate a single SpatialNeuron is more than the equivalent memory cost of a pool of 19000 “Dendrify neurons”. Of course, we do **not** interpret this result as *Dendrify being 19000 times more efficient than SpatialNeuron*. In contrast to execution times, memory needs are harder to gauge accurately and sometimes are irrelevant to real-world performance; if more memory results in better performance or increased accuracy, then high memory need is not necessarily bad. However, these results point towards our equation-based approach being significantly more efficient than a SpatialNeuron-based approach.

Conclusion 2: Based on the implementation constraints explained above and the benchmarks and memory test results, we conclude that using SpatialNeuron for network simulations in Brian 2 is currently a suboptimal and inefficient solution. Dendriify is more compatible with Brian’s design philosophy, offering significant performance advantages than the built-in SpatialNeuron when running simulations of reduced compartmental neurons at a large scale.

Action 3: We tested if using explicit integration methods results in simulation errors and numerical instabilities. Biophysically and morphologically detailed models are very stiff and require complex implicit methods. However, with Dendriify, we aim to extend the “point-neuron” idea by adding a few compartments that account for specific regions in the dendritic morphology. Thus, our approach typically results in reduced compartmental neuron models that share these characteristics:

1. They have small compartments (usually around 3-5).
2. Each compartment can be quite long (>100 μm).
3. Model compartments are not divided into segments; thus, the number of segments equals the number of compartments.

As explained in “Numerical Methods for Neuronal Modeling.” Methods in Neuronal Modeling 2 (1998) attached by the reviewer: *“The stiffness gets extremely large as we increase the number of compartments in our cable model. In fact, it grows as the square of the number of compartments in our models. Thus we see that compartmental models can be very stiff when there is little dissipation via the membrane conductance. Paradoxically this stiffness increases as we use smaller and smaller compartments to better resolve spatial details.”*

Since our approach utilizes neurons with a small number of big compartments, we expect that explicit approaches and a reasonable simulation time step would not cause any substantial numerical issues. To test this hypothesis, we directly compared Dendriify against SpatialNeuron (which utilizes an implicit method) using the same 4-compartment model as before and a challenging simulation protocol. Notably, the comparison between the two approaches was also suggested by the Brian team. Their prediction was that: *“... for small dt , the results are very similar, but at some point, the dendriify simulation will do something completely wrong (or generate NaN or something like that), while SpatialNeuron will become inaccurate in a smooth way.”*

Test details:

- A very high frequency (300 Hz) Poisson input is provided at the most distal dendritic compartment.
- This input generates synaptic currents of fast kinetics (instant rise and 2 ms decay time constant).
- The synaptic weight is large enough to cause robust somatic activation (~ 8 Hz). Typically, inputs to distal branches of pyramidal neurons fail to do that.
- Simulation time step: Ranged from 0.025 ms to 0.425 ms (with step 0.025 ms).
- We tested all integration methods that are available in Brian 2.

Results summary:

1. When $dt \leq 0.1$ ms, a “Dendriify neuron” generates voltage responses and somatic spike times nearly identical to a SpatialNeuron, regardless of the integration method used.
2. When $dt > 0.1$ ms, we observe deviations between Dendriify and SpatialNeuron that affect the somatic spike times and less the voltage traces.
3. When $dt = 0.425$ ms, both approaches become unstable and fail. We also confirm the Brian team’s prediction that *“the dendriify simulation will do something completely wrong (or generate NaN or something like that), while SpatialNeuron will become inaccurate in a smooth way”*. However, this is true only for the

forward Euler method; all the other integration methods are numerically stable and comparable to a SpatialNeuron.

Important notes

1. According to Brian's documentation, an error can be introduced during a simulation because spike times are constrained to a grid and cannot occur at arbitrary times. "Note that the inaccuracy introduced by the spike time approximation is of order $O(dt)$, so the total accuracy of the simulation is of order $O(dt)$ per time step. This means that regardless of the integration method, increasing the dt also increases the error linked to how spike times are calculated. It is essential to keep this in mind when comparing Dendrify and SpatialNeuron, especially when the dt is high.
2. For this comparison we run hundreds of different combinations of integration method, dt and simulation time. For practical reasons, we provide below only the data for $dt \in \{0.05, 0.1, 0.15\}$ and method $\in \{\text{Forward Euler, Exponential Euler, 2}^{\text{nd}} \text{ order Runge-Kutta, 4}^{\text{th}} \text{ order Runge-Kutta, Heun's method}\}$.
3. We color coded the different simulation time steps (yellow = 0.05 ms, green = 0.1 ms, blue = 0.15 ms)

dt = 0.05 ms

300 Hz synaptic input to Dendrite 3
(dt = 0.050 ms | method = exponential_euler)

300 Hz synaptic input to Dendrite 3
(dt = 0.050 ms | method = rk2)

dt = 0.1 ms

300 Hz synaptic input to Dendrite 3
(dt = 0.100 ms | method = rk2)

dt = 0.15 ms

300 Hz synaptic input to Dendrite 3
(dt = 0.150 ms | method = euler)

300 Hz synaptic input to Dendrite 3
(dt = 0.150 ms | method = exponential_euler)

300 Hz synaptic input to Dendrite 3
(dt = 0.150 ms | method = rk2)

Conclusion 3: We observe that for a small number of compartments and with $dt \leq 0.1$ ms, our approach is numerically stable and has almost identical results with Brian's SpatialNeuron which utilizes an implicit integration method for solving model equations.

Action 4: We updated the manuscript's Discussion (Pages: 30-31, Lines: 437-448) and DendriFY's documentation to highlight the limitations of our approach. We also added some of the abovementioned tests to the Supplement (Figures S8-S17 | Simulation accuracy and numerical stability analysis) for the interested reader.

C) Software best practices

The Python code looks reasonable in form and style. However, you need to implement the following:

- * continuous integration and tests (also against upstream Brian2)*
- * continuous delivery, e.g. provide the latest release on PyPI*
- * linting and style checking*
- * published documentation, e.g. readthedocs.org*

You and your users want to use the tool for many years. But without the points above, it will degrade. Especially the checks against upstream Brian2 are vital.

We thank the reviewer for this vital point! We have taken the following actions to comply with their suggestions.

Actions taken:

1. We debugged, improved, and formatted (using `autopep8`) the entire DendriFY code.
2. We documented every public class, property, method, or parameter. We also included type hints and links to third-party documentation when necessary.
3. We created a new, official GitHub repository: <https://github.com/Poirazi-Lab/dendriFY>
4. We also created a new documentation website (using sphinx): <https://dendriFY.readthedocs.io/en/latest/>. This website is automatically updated every time we push commits to the main branch. New content such as code examples and tutorials, will be added soon. All Jupyter notebooks that reproduce the manuscript's figures have been transferred there as well.
5. DendriFY can now be easily installed through PyPI: <https://pypi.org/project/dendriFY/>. The latest package version is automatically uploaded with every new GitHub release.

Regarding continuous testing, we are working on it, but we have not completed it yet. However, achieving good test coverage will be one of our top priorities in the immediate future. Currently, DendriFY works perfectly with the latest Brian 2 version.

D) Performance

Please show the performance of your implementation. How long does it take to simulate one neuron, 10, 1000? What about networks of neurons?

We thank the reviewer for another great suggestion. First, as the reviewer is certainly aware of, simulation performance depends on multiple factors, including the selected Brian operation mode, model complexity, and

hardware specifications. Additionally, model optimization usually occurs at the late stages of development, if necessary, and also depends on a user's specific needs. It is also worth mentioning that there are currently two different approaches that make Brian 2 run on Nvidia GPUs, namely Brian2GeNN and Brian2CUDA, with the latter also being compatible with Dendriify (verified by its developers, but we have not tested it yet). Thus, finding the single most representative use case to test the performance of our implementation is not so straightforward.

Having said that, we designed a series of simple benchmarks to test Dendriify under some realistic conditions and have added a new section (**Scalability Analysis**, Pages: 24-27, Lines: 322-357) and a new figure (figure **6**, also included below) in the manuscript that exemplifies our findings.

Benchmark details

Neuron model:

- 4 compartments (3 dendrites + leaky IF soma)
- External input -> Poisson generators
- Synapses with AMPA-like kinetics

Simulation details:

- Simulated time -> 1 second
- dt = 0.1 ms
- Integration method: Forward Euler

System details:

- Ubuntu 22.04.1 laptop
- i7-9750H CPU
- 16 GB RAM

Setup:

- Jupyter notebook (%%timeit module)
- Run Brian in "numpy" mode
- Measured model build + runtime (mean of 10 runs)

Tests cases:

1. A group of N neurons with **passive** dendrites, 2 N generators, and **no interneuronal connections**
2. A group of N neurons with **active** dendrites, 2 N generators, and **no interneuronal connections**
3. A group of N neurons with **active** dendrites, 2 N generators with **recurrent connections** (~50 synapses/neuron)

Figure 6 | Estimating DendriFY's performance when increasing network complexity & size. a) Schematic illustration of the three model cases used for the benchmark tests. In all cases, the neuronal model was an adapted version of the 4-compartment model shown in **Fig. 2a**. Note that the number of Poisson input generators scaled with N . Left: a group of N neurons with passive dendrites and no recurrent synapses. Middle: a group of N neurons with active dendrites (i.e. furnished with Na^+ spikes) and no recurrent synapses. Right: a recurrent network of N neurons with active dendrites and ~ 50 synapses/neuron. **b)** The benchmark results, showing how the combined build and simulation time scales when increasing N . The times plotted here represent the average of 10 runs. Simulations were performed on a laptop (blue, orange, green) or an iPad (black). For more information, refer to **table S4**. All benchmark codes and the raw results are available on GitHub.

Minor points

1) You stress that Brian2 has seamless compatibility with Dendripy which is a bit funny as Dendripy is based on Brian2 and not the other way around.

The reviewer is right; we have removed any references to “seamless compatibility” in the text to prevent any misunderstanding.

2) All examples work with single neurons although you often, especially in the introduction, talk about networks. Please add at least one example of a network of multi-compartment neurons.

This is a valid point. In response to this comment, we have added a new figure (**Figure 6**) in the manuscript showing the performance of increasing numbers of neurons with passive and active dendrites stimulated with synaptic input. This figure consists of both isolated (groups) of neurons as well as interconnected (networks) of neurons.

3) “The attenuation of currents traveling along the somatodendritic axis is an intrinsic property of biological neurons and is due to the morphology and cable properties of dendritic trees.”

You don't really show currents traveling along the axis as you have only a handful of compartments.

Thank you. This comment was not referring to the toy model but rather to the signal attenuation seen in biological neurons. We have rephrased the sentence that refers to the model to clarify this (Page: 7, Lines: 116-117).

4) neuromorphic implementations

Please cite more multi-compartment neuromorphic implementations, e.g. “Emulating Dendritic Computing Paradigms on Analog Neuromorphic Hardware” Kaiser et al. <https://doi.org/10.1016/j.neuroscience.2021.08.013>

Thank you for the recommendation. This is indeed a very interesting study that we are familiar with and was an oversight that it was not included in the original manuscript. It is now added.

Thanks,
your reviewer

Thank you!!

Reviewer #2

The authors introduce a python-package to facilitate the modeling of dendritic compartments in spiking networks. The objective is an easy to use and computationally efficient framework which captures dendritic features currently omitted in the simulation packages.

On the basis of developments in neurophysiology, computational neuroscience, and recent results in the field of machine learning, they outline how accounting for dendritic phenomenon is necessary to facilitate advancements in computational neuroscience and neuromorphic computing.

The authors set up four model scenarios to exemplify dendritic behaviour.

- * Coupling of passive dendrites
- * Compartmental model with active dendrites
- * A compartmental model fit to a CA1 pyramidal cell.
- * Coincidence detection in populations of CA1 model neurons.

The code is very well documented and accessible at their github repository. This will make it very easy to use for anyone with some programming experience.

Lastly, the authors example a practical guide of developing models with their package.

Developing joint, open source software is extremely important to facilitate research progress and to validate results. Here, this package fills an important gap.

We did check the code and did not find technical problems.

The manuscript is well written and well understandable, serving as a good guide for starters.

We are grateful to the reviewer for their positive feedback and appreciation of our work!

Comments

The authors mention several times the computational capabilities, and the compatibility of dendrify with brian 2, yet are not clearly mentioning any of the specifics.

Their github actually makes clear that their implementation is fully build on top of the existing brian 2 frameworks. Which is appropriate as everything can be reinterpreted as more complicated neuron models. This should be clarified in the manuscript as well.

We thank the reviewer for this important comment. We have now clarified in various places in the text that Dendrify builds on top of Brian 2. We have also added an extensive set of comparisons in the **Discussion** (Pages: 29-30, Lines: 403-422) that refers to other simulators and points out the differences in their approach vs. ours. We also directly compared our approach to the SpatialNeuron class of Brian 2 (see: **Figures S8-S17 | Simulation accuracy and numerical stability analysis**). For more advantages of Dendrify over SpatialNeuron, please refer to our response to Reviewer 1, comment B.

On a technical note, the formatting of the manuscript make any mathematical derivations hard to read, and we hope that the next version improves here.

We thank the reviewer for this comment. We are unsure as to exactly what the reviewer refers to, however we have revised the derivation of the coupling conductances and improved formatting in various Method sections (Lines: 527-530, 535-539, 548-549, 594-597). If you have any specific recommendations on the notation or formatting of the equations, please feel free to share them with us. All remaining equations are taken from the following books^{1,2}.

References

1. Ermentrout, G. B. & Terman, D. H. Mathematical foundations of neuroscience. in *Interdisciplinary Applied Mathematics* (2010).
2. Chavlis, S. & Poirazi, P. Modeling Dendrites and Spatially-Distributed Neuronal Membrane Properties. in *Computational Modelling of the Brain: Modelling Approaches to Cells, Circuits and Networks* (eds. Giugliano, M., Negrello, M. & Linaro, D.) 25–67 (Springer International Publishing, 2022). doi:10.1007/978-3-030-89439-9_2.

REVIEWERS' COMMENTS

Reviewer #1 (Remarks to the Author):

Dear authors!

Thanks for revising the manuscript. You've covered all my points. Best, your reviewer.

Reviewer #2 (Remarks to the Author):

The authors have addressed my points.

It would be great if already in the abstract it would be clearly mentioned that their toolbox is fully built on Brian 2.